# Transferring Knowledge
# across Learning Processes

**Sebastian Flennerhag**[*]
The Alan Turing Institute
London, UK
`sflennerhag@turing.ac.uk`

**Pablo G. Moreno**
Amazon
Cambridge, UK
`morepabl@amazon.com`

**Neil D. Lawrence**
Amazon
Cambridge, UK
`lawrennd@amazon.com`

**Andreas Damianou**
Amazon
Cambridge, UK
`damianou@amazon.com`

## ABSTRACT

In complex transfer learning scenarios new tasks might not be tightly linked to previous tasks. Approaches that transfer information contained only in the final parameters of a source model will therefore struggle. Instead, transfer learning at a higher level of abstraction is needed. We propose Leap, a framework that achieves this by transferring knowledge across learning processes. We associate each task with a manifold on which the training process travels from initialization to final parameters and construct a meta-learning objective that minimizes the expected length of this path. Our framework leverages only information obtained during training and can be computed on the fly at negligible cost. We demonstrate that our framework outperforms competing methods, both in meta-learning and transfer learning, on a set of computer vision tasks. Finally, we demonstrate that Leap can transfer knowledge across learning processes in demanding reinforcement learning environments (Atari) that involve millions of gradient steps.

## 1 INTRODUCTION

Transfer learning is the process of transferring knowledge encoded in one model trained on one set of tasks to another model that is applied to a new task. Since a trained model encodes information in its learned parameters, transfer learning typically transfers knowledge by encouraging the target model's parameters to resemble those of a previous (set of) model(s) (Pan & Yang, 2009). This approach limits transfer learning to settings where good parameters for a new task can be found in the neighborhood of parameters that were learned from a previous task. For this to be a viable assumption, the two tasks must have a high degree of structural affinity, such as when a new task can be learned by extracting features from a pretrained model (Girshick et al., 2014; He et al., 2017; Mahajan et al., 2018). If not, this approach has been observed to limit knowledge transfer since the training process on one task will discard information that was irrelevant for the task at hand, but that would be relevant for another task (Higgins et al., 2017; Achille et al., 2018).

We argue that such information can be harnessed, even when the downstream task is unknown, by transferring knowledge of the learning process itself. In particular, we propose a meta-learning framework for aggregating information across task geometries as they are observed during training. These geometries, formalized as the loss surface, encode all information seen during training and thus avoid catastrophic information loss. Moreover, by transferring knowledge across learning processes, information from previous tasks is distilled to explicitly facilitate the learning of new tasks.

Meta learning frames the learning of a new task as a learning problem itself, typically in the few-shot learning paradigm (Lake et al., 2011; Santoro et al., 2016; Vinyals et al., 2016). In this

---

[*]Work done while at Amazon.

environment, learning is a problem of rapid adaptation and can be solved by training a meta-learner by backpropagating through the entire training process (Ravi & Larochelle, 2016; Andrychowicz et al., 2016; Finn et al., 2017). For more demanding tasks, meta-learning in this manner is challenging; backpropagating through thousands of gradient steps is both impractical and susceptible to instability. On the other hand, truncating backpropagation to a few initial steps induces a short-horizon bias (Wu et al., 2018). We argue that as the training process grows longer in terms of the distance traversed on the loss landscape, the geometry of this landscape grows increasingly important. When adapting to a new task through a single or a handful of gradient steps, the geometry can largely be ignored. In contrast, with more gradient steps, it is the dominant feature of the training process.

To scale meta-learning beyond few-shot learning, we propose *Leap*, a light-weight framework for meta-learning over task manifolds that does not need any forward- or backward-passes beyond those already performed by the underlying training process. We demonstrate empirically that Leap is a superior method to similar meta and transfer learning methods when learning a task requires more than a handful of training steps. Finally, we evaluate Leap in a reinforcement Learning environment (Atari 2600; Bellemare et al., 2013), demonstrating that it can transfer knowledge across learning processes that require millions of gradient steps to converge.

## 2 Transferring Knowledge across Learning Processes

We start in section 2.1 by introducing the gradient descent algorithm from a geometric perspective. Section 2.2 builds a framework for transfer learning and explains how we can leverage geometrical quantities to transfer knowledge across learning processes by guiding gradient descent. We focus on the point of initialization for simplicity, but our framework can readily be extended. Section 2.3 presents Leap, our lightweight algorithm for transfer learning across learning processes.

### 2.1 Gradient Paths on Task Manifolds

Central to our framework is the notion of a learning process; the harder a task is to learn, the harder it is for the learning process to navigate on the loss surface (fig. 1). Our framework is based on the idea that transfer learning can be achieved by leveraging information contained in similar learning processes. Exploiting that this information is encoded in the geometry of the loss surface, we leverage geometrical quantities to facilitate the learning process with respect to new tasks. We focus on the supervised learning setting for simplicity, though our framework applies more generally. Given a learning objective $f$ that consumes an input $x \in \mathbb{R}^m$ and a target $y \in \mathbb{R}^c$ and maps a parameterization $\theta \in \mathbb{R}^n$ to a scalar loss value, we have the gradient descent update as

$$\theta^{i+1} = \theta^i - \alpha^i S^i \nabla f(\theta^i), \tag{1}$$

where $\nabla f(\theta^i) = \mathbb{E}_{x,y \sim p(x,y)} \big[ \nabla f(\theta^i, x, y) \big]$. We take the learning rate schedule $\{\alpha^i\}_i$ and preconditioning matrices $\{S^i\}_i$ as given, but our framework can be extended to learn these jointly with the initialization. Different schemes represent different optimizers; for instance $\alpha^i = \alpha$, $S^i = I_n$ yields gradient descent, while defining $S^i$ as the inverse Fisher matrix results in natural gradient descent (Amari, 1998). We assume this process converges to a stationary point after $K$ gradient steps.

To distinguish different learning processes originating from the same initialization, we need a notion of their length. The longer the process, the worse the initialization is (conditional on reaching equivalent performance, discussed further below). Measuring the Euclidean distance between initialization and final parameters is misleading as it ignores the actual path taken. This becomes crucial when we compare paths from different tasks, as gradient paths from different tasks can originate from the same initialization and converge to similar final parameters, but take very different paths. Therefore, to capture the length of a learning process we must associate it with the loss surface it traversed.

The process of learning a task can be seen as a curve on a specific task manifold $M$. While this manifold can be constructed in a variety of ways, here we exploit that, by definition, any learning process traverses the loss surface of $f$. As such, to accurately describe the length of a gradient-based learning process, it is sufficient to define the task manifold as the loss surface. In particular, because the learning process in eq. 1 follows the gradient trajectory, it constantly provides information about the

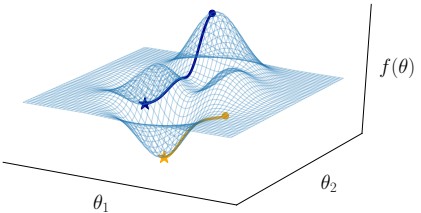

Figure 1: Example of gradient paths on a manifold described by the loss surface. Leap learns an initialization with shorter expected gradient path that improves performance.

geometry of the loss surface. Gradients that largely point in the same direction indicate a well-behaved loss surface, whereas gradients with frequently opposing directions indicate an ill-conditioned loss surface—something we would like to avoid. Leveraging this insight, we propose a framework for transfer learning that exploits the accumulation of geometric information by constructing a meta objective that minimizes the expected length of the gradient descent path *across tasks*. In doing so, the meta objective intrinsically balances local geometries across tasks and encourages an initialization that makes the learning process as short as possible.

To formalize the notion of the distance of a learning process, we define a task manifold $M$ as a submanifold of $\mathbb{R}^{n+1}$ given by the graph of $f$. Every point $p = (\theta, f(\theta)) \in M$ is locally homeomorphic to a Euclidean subspace, described by the tangent space $T_p M$. Taking $\mathbb{R}^{n+1}$ to be Euclidean, it is a Riemann manifold. By virtue of being a submanifold of $\mathbb{R}^{n+1}$, $M$ is also a Riemann manifold. As such, $M$ comes equipped with an smoothly varying inner product $g_p : T_p M \times T_p M \mapsto \mathbb{R}$ on tangent spaces, allowing us to measure the length of a path on $M$. In particular, the length (or energy) of any *curve* $\gamma : [0, 1] \mapsto M$ is defined by accumulating infinitesimal changes along the trajectory,

$$\text{Length}(\gamma) = \int_0^1 \sqrt{g_{\gamma(t)}(\dot{\gamma}(t), \dot{\gamma}(t))}\, dt, \qquad \text{Energy}(\gamma) = \int_0^1 g_{\gamma(t)}(\dot{\gamma}(t), \dot{\gamma}(t))\, dt, \qquad (2)$$

where $\dot{\gamma}(t) = \frac{d}{dt}\gamma(t) \in T_{\gamma(t)}M$ is a tangent vector of $\gamma(t) = (\theta(t), f(\theta(t))) \in M$. We use parentheses (i.e. $\gamma(t)$) to differentiate discrete and continuous domains. With $M$ being a submanifold of $\mathbb{R}^{n+1}$, the induced metric on $M$ is defined by $g_{\gamma(t)}(\dot{\gamma}(t), \dot{\gamma}(t)) = \langle \dot{\gamma}(t), \dot{\gamma}(t) \rangle$. Different constructions of $M$ yield different Riemann metrics. In particular, if the model underlying $f$ admits a predictive probability distribution $P(y \mid x)$, the task manifold can be given an information geometric interpretation by choosing the Fisher matrix as Riemann metric, in which case the task manifold is defined over the space of probability distributions (Amari & Nagaoka, 2007). If eq. 1 is defined as natural gradient descent, the learning process corresponds to gradient descent on this manifold (Amari, 1998; Martens, 2010; Pascanu & Bengio, 2014; Luk & Grosse, 2018).

Having a complete description of a task manifold, we can measure the length of a learning process by noting that gradient descent can be seen as a discrete approximation to the scaled gradient flow $\dot{\theta}(t) = -S(t)\nabla f(\theta(t))$. This flow describes a curve that originates in $\gamma(0) = (\theta^0, f(\theta^0))$ and follows the gradient at each point. Going forward, we define $\gamma$ to be this unique curve and refer to it as the *gradient path* from $\theta^0$ on $M$. The metrics in eq. 2 can be computed exactly, but in practice we observe a discrete learning process. Analogously to how the gradient update rule approximates the gradient flow, the gradient path length or energy can be approximated by the cumulative chordal distance (Ahlberg et al., 1967),

$$d_p(\theta^0, M) = \sum_{i=0}^{K-1} \|\gamma^{i+1} - \gamma^i\|_2^p, \qquad p \in \{1, 2\}. \qquad (3)$$

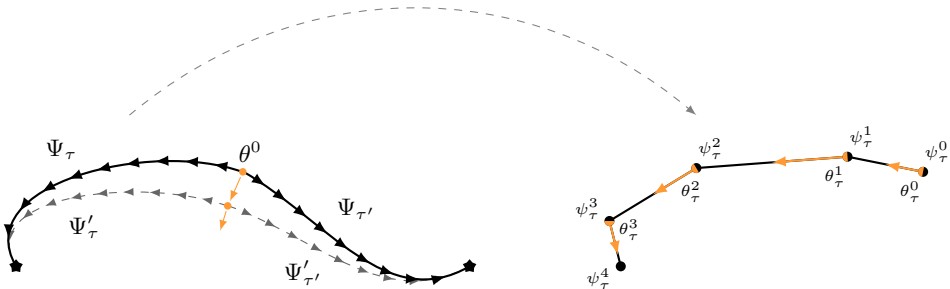

Figure 2: *Left:* illustration of Leap (algorithm 1) for two tasks, $\tau$ and $\tau'$. From an initialization $\theta^0$, the learning process of each task generates gradient paths, $\Psi_\tau$ and $\Psi_{\tau'}$, which Leap uses to minimize the expected path length. Iterating the process, Leap converges to a locally Pareto optimal initialization. *Right:* the pull-forward objective (eq. 6) used to minimize the expected gradient path length. Any gradient path $\Psi_\tau = \{\psi_\tau^i\}_{i=1}^{K_\tau}$ acts on $\theta^0$ by pulling each $\theta_\tau^i$ towards $\psi_\tau^{i+1}$.

We write $d$ when the distinction between the length or energy metric is immaterial. Using the energy yields a slightly simpler objective, but the length normalizes each length segment and as such protects against differences in scale between task objectives. In appendix C, we conduct an ablation study and find that they perform similarly, though using the length leads to faster convergence. Importantly, $d$ involves only terms seen during task training. We exploit this later when we construct the meta gradient, enabling us to perform gradient descent on the meta objective at negligible cost (eq. 8).

We now turn to the transfer learning setting where we face a set of tasks, each with a distinct task manifold. Our framework is built on the idea that we can transfer knowledge across learning processes via the local geometry by aggregating information obtained along observed gradient paths. As such, Leap finds an initialization from which learning converges as rapidly as possible in expectation.

## 2.2 META LEARNING ACROSS TASK MANIFOLDS

Formally, we define a task $\tau = (f_\tau, p_\tau, u_\tau)$ as the process of learning to approximate the relationship $x \mapsto y$ through samples from the data distribution $p_\tau(x, y)$. This process is defined by the gradient update rule $u_\tau$ (as defined in eq. 1), applied $K_\tau$ times to minimize the task objective $f_\tau$. Thus, a learning process starts at $\theta_\tau^0 = \theta^0$ and progresses via $\theta_\tau^{i+1} = u_\tau(\theta_\tau^i)$ until $\theta_\tau^{K_\tau}$ is obtained. The sequence $\{\theta_\tau^i\}_{i=0}^{K_\tau}$ defines an approximate gradient path on the task manifold $M_\tau$ with distance $d(\theta^0; M_\tau)$.

To understand how $d$ transfers knowledge across learning processes, consider two distinct tasks. We can transfer knowledge across these tasks' learning processes by measuring how good a shared initialization is. Assuming two candidate initializations converge to limit points with equivalent performance on each task, the initialization with shortest expected gradient path distance encodes more knowledge sharing. In particular, if both tasks have convex loss surfaces a unique optimal initialization exists that achieves Pareto optimality in terms of total path distance. This can be crucial in data sparse regimes: rapid convergence may be the difference between learning a task and failing due to overfitting (Finn et al., 2017).

Given a distribution of tasks $p(\tau)$, each candidate initialization $\theta^0$ is associated with a measure of its expected gradient path distance, $\mathbb{E}_{\tau \sim p(\tau)}\big[d(\theta^0; M_\tau)\big]$, that summarizes the suitability of the initialization to the task distribution. The initialization (or a set thereof) with shortest expected gradient path distance maximally transfers knowledge across learning processes and is Pareto optimal in this regard. Above, we have assumed that all candidate initializations converge to limit points of equal performance. If the task objective $f_\tau$ is non-convex this is not a trivial assumption and the gradient path distance itself does not differentiate between different levels of final performance.

As such, it is necessary to introduce a feasibility constraint to ensure only initializations with some minimum level of performance are considered. We leverage that transfer learning never happens in a vacuum; we always have a second-best option, such as starting from a random initialization or a pretrained model. This "second-best" initialization, $\psi^0$, provides us with the performance we

---

**Algorithm 1** Leap: Transferring Knowledge over Learning Processes

---

**Require:** $p(\tau)$, $\tau = (f_\tau, u_\tau, p_\tau)$: distribution over tasks
**Require:** $\beta$: step size
 1: randomly initialize $\theta^0$
 2: **while** not done **do**
 3:     $\nabla \bar{F} \leftarrow 0$: initialize meta gradient
 4:     sample task batch $\mathcal{B}$ from $p(\tau)$
 5:     **for all** $\tau \in \mathcal{B}$ **do**
 6:         $\psi_\tau^0 \leftarrow \theta^0$: initialize task baseline
 7:         **for all** $i \in \{0, \dots, K_\tau - 1\}$ **do**
 8:             $\psi_\tau^{i+1} \leftarrow u_\tau(\psi_\tau^i)$: update baseline
 9:             $\theta_\tau^i \leftarrow \psi_\tau^i$: follow baseline (recall $\psi_\tau^0 = \theta^0$)
10:             increment $\nabla \bar{F}$ using the pull-forward gradient (eq. 8)
11:         **end for**
12:     **end for**
13:     $\theta^0 \leftarrow \theta^0 - \frac{\beta}{|\mathcal{B}|} \nabla \bar{F}$: update initialization
14: **end while**

---

would obtain on a given task in the absence of knowledge transfer. As such, performance obtained by initializing from $\psi^0$ provides us with an upper bound for each task: a candidate solution $\theta^0$ must achieve at least as good performance to be a viable solution. Formally, this implies the task-specific requirement that a candidate $\theta^0$ must satisfy $f_\tau(\theta_\tau^{K_\tau}) \leq f_\tau(\psi_\tau^{K_\tau})$. As this must hold for every task, we obtain the canonical meta objective

$$
\begin{aligned}
\min_{\theta^0} \quad & F(\theta^0) = \mathbb{E}_{\tau \sim p(\tau)} \big[ d(\theta^0; M_\tau) \big] \\
\text{s.t.} \quad & \theta_\tau^{i+1} = u_\tau(\theta_\tau^i), \quad \theta_\tau^0 = \theta^0, \\
& \theta^0 \in \Theta = \cap_\tau \big\{ \theta^0 \mid f_\tau(\theta_\tau^{K_\tau}) \leq f_\tau(\psi_\tau^{K_\tau}) \big\}.
\end{aligned}
\tag{4}
$$

This meta objective is robust to variations in the geometry of loss surfaces, as it balances complementary and competing learning processes (fig. 2). For instance, there may be an initialization that can solve a small subset of tasks in a handful of gradient steps, but would be catastrophic for other related tasks. When transferring knowledge via the initialization, we must trade off commonalities and differences between gradient paths. In eq. 4 these trade-offs arise naturally. For instance, as the number of tasks whose gradient paths move in the same direction increases, so does their pull on the initialization. Conversely, as the updates to the initialization renders some gradient paths longer, these act as springs that exert increasingly strong pressure on the initialization. The solution to eq. 4 thus achieves an equilibrium between these competing forces.

Solving eq. 4 naively requires training to convergence on each task to determine whether an initialization satisfies the feasibility constraint, which can be very costly. Fortunately, because we have access to a second-best initialization, we can solve eq. 4 more efficiently by obtaining gradient paths from $\psi^0$ and use these as baselines that we incrementally improve upon. This improved initialization converges to the same limit points, but with shorter expected gradient paths (theorem 1). As such, it becomes the new second-best option; Leap (algorithm 1) repeats this process of improving upon increasingly demanding baselines, ultimately finding a solution to the canonical meta objective.

### 2.3 LEAP

Leap starts from a given second-best initialization $\psi^0$, shared across all tasks, and constructs baseline gradient paths $\Psi_\tau = \{\psi_\tau^i\}_{i=0}^{K_\tau}$ for each task $\tau$ in a batch $\mathcal{B}$. These provide a set of baselines $\Psi = \{\Psi_\tau\}_{\tau \in \mathcal{B}}$. Recall that all tasks share the same initialization, $\psi_\tau^0 = \psi^0 \in \Theta$. We use these baselines, corresponding to task-specific learning processes, to modify the gradient path distance metric in eq. 3 by freezing the forward point $\gamma_\tau^{i+1}$ in all norms,

$$\bar{d}_p(\theta^0; M_\tau, \Psi_\tau) = \sum_{i=0}^{K_\tau - 1} \|\bar{\gamma}_\tau^{i+1} - \gamma_\tau^i\|_2^p, \tag{5}$$

where $\bar{\gamma}_\tau^i = (\psi_\tau^i, f(\psi_\tau^i))$ represents the frozen forward point from the baseline and $\gamma_\tau^i = (\theta_\tau^i, f(\theta_\tau^i))$ the point on the gradient path originating from $\theta^0$. This surrogate distance metric encodes the feasibility constraint; optimizing $\theta^0$ with respect to $\Psi$ pulls the initialization forward along each task-specific gradient path in an unconstrained variant of eq. 4 that replaces $\Theta$ with $\Psi$,

$$\begin{aligned} \min_{\theta^0} \quad & \bar{F}(\theta^0; \Psi) = \mathbb{E}_{\tau \sim p(\tau)}\left[\bar{d}(\theta^0; M_\tau, \Psi_\tau)\right], \\ \text{s.t.} \quad & \theta_\tau^{i+1} = u_\tau(\theta_\tau^i), \quad \theta_\tau^0 = \theta^0. \end{aligned} \tag{6}$$

We refer to eq. 6 as the *pull-forward* objective. Incrementally improving $\theta^0$ over $\psi^0$ leads to a new second-best option that Leap uses to generate a new set of more demanding baselines, to further improve the initialization. Iterating this process, Leap produces a sequence of candidate solutions to eq. 4, all in $\Theta$, with incrementally shorter gradient paths. While the pull-forward objective can be solved with any optimization algorithm, we consider gradient-based methods. In theorem 1, we show that gradient descent on $\bar{F}$ yields solutions that always lie in $\Theta$. In principle, $\bar{F}$ can be evaluated at any $\theta^0$, but a more efficient strategy is to evaluate $\theta^0$ at $\psi^0$. In this case, $\bar{d} = d$, so that $\bar{F} = F$.

**Theorem 1** (Pull-forward). *Define a sequence of initializations $\{\theta_s^0\}_{s \in \mathbb{N}}$ by*

$$\theta_{s+1}^0 = \theta_s^0 - \beta_s \nabla \bar{F}(\theta_s^0; \Psi_s), \qquad \theta^0 \in \Theta, \tag{7}$$

*with $\psi_s^0 = \theta_s^0$ for all $s$. For $\beta_s > 0$ sufficiently small, there exist learning rates schedules $\{\alpha_\tau^i\}_{i=1}^{K_\tau}$ for all tasks such that $\theta_{k \to \infty}^0$ is a limit point in $\Theta$.*

Proof: see appendix A. Because the meta gradient requires differentiating the learning process, we must adopt an approximation. In doing so, we obtain a meta-gradient that can be computed analytically on the fly during task training. Differentiating $\bar{F}$, we have

$$\nabla \bar{F}(\theta^0, \Psi) = -p\, \mathbb{E}_{\tau \sim p(\tau)}\left[\sum_{i=0}^{K_\tau - 1} J_\tau^i(\theta_\tau^0)^T \left(\Delta f_\tau^i \nabla f_\tau(\theta_\tau^i) + \Delta \theta_\tau^i\right)\left(\|\bar{\gamma}_\tau^{i+1} - \gamma_\tau^i\|_2^p\right)^{p-2}\right] \tag{8}$$

where $J_\tau^i$ denotes the Jacobian of $\theta_\tau^i$ with respect to the initialization, $\Delta f_\tau^i = f_\tau(\psi_\tau^{i+1}) - f_\tau(\theta_\tau^i)$ and $\Delta \theta_\tau^i = \psi_\tau^{i+1} - \theta_\tau^i$. To render the meta gradient tractable, we need to approximate the Jacobians, as these are costly to compute. Empirical evidence suggest that they are largely redundant (Finn et al., 2017; Nichol et al., 2018). Nichol et al. (2018) further shows that an identity approximation yields a meta-gradient that remains faithful to the original meta objective. We provide some further support for this approximation (see appendix B). First, we note that the learning rate directly controls the quality of the approximation; for any $K_\tau$, the identity approximation can be made arbitrarily accurate by choosing a sufficiently small learning rates. We conduct an ablation study to ascertain how severe this limitation is and find that it is relatively loose. For the best-performing learning rate, the identity approximation is accurate to four decimal places and shows no signs of significant deterioration as the number of training steps increases. As such, we assume $J^i \approx I_n$ throughout. Finally, by evaluating $\nabla \bar{F}$ at $\theta^0 = \psi^0$, the meta gradient contains only terms seen during standard training and can be computed asynchronously on the fly at negligible cost.

In practice, we use stochastic gradient descent during task training. This injects noise in $f$ as well as in its gradient, resulting in a noisy gradient path. Noise in the gradient path does not prevent Leap from converging. However, noise reduces the rate of convergence, in particular when a noisy gradient step results in $f_\tau(\psi_\tau^{s+1}) - f_\tau(\theta_\tau^i) > 0$. If the gradient estimator is reasonably accurate, this causes the term $\Delta f_\tau^i \nabla f_\tau(\theta_\tau^i)$ in eq. 8 to point in the steepest ascent direction. We found that adding a stabilizer to ensure we always follow the descent direction significantly speeds up convergence and allows us to use larger learning rates. In this paper, we augment $\bar{F}$ with a stabilizer of the form

$$\mu\left(f_\tau(\theta_\tau^i); f_\tau(\psi_\tau^{i+1})\right) = \begin{cases} 0 & \text{if} \quad f_\tau(\psi_\tau^{i+1}) \le f_\tau(\theta_\tau^i), \\ -2(f_\tau(\psi_\tau^{i+1}) - f_\tau(\theta_\tau^i))^2 & \text{else.} \end{cases}$$

Adding $\nabla\mu$ (re-scaled if necessary) to the meta-gradient is equivalent to replacing $\Delta f_\tau^i$ with $-|\Delta f_\tau^i|$ in eq. 8. This ensures that we never follow $\nabla f_\tau(\theta_\tau^i)$ in the ascent direction, instead reinforcing the descent direction at that point. This stabilizer is a heuristic, there are many others that could prove helpful. In appendix C we perform an ablation study and find that the stabilizer is not necessary for Leap to converge, but it does speed up convergence significantly.

## 3 RELATED WORK

Transfer learning has been explored in a variety of settings, the most typical approach attempting to infuse knowledge in a target model's parameters by encouraging them to lie close to those of a pretrained source model (Pan & Yang, 2009). Because such approaches can limit knowledge transfer (Higgins et al., 2017; Achille et al., 2018), applying standard transfer learning techniques leads to *catastrophic forgetting*, by which the model is rendered unable to perform a previously mastered task (McCloskey & Cohen, 1989; Goodfellow et al., 2013). These problems are further accentuated when there is a larger degree of diversity among tasks that push optimal parameterizations further apart. In these cases, transfer learning can in fact be worse than training from scratch.

Recent approaches extend standard finetuning by adding regularizing terms to the training objective that encourage the model to learn parameters that both solve a new task and retain high performance on previous tasks. These regularizers operate by protecting the parameters that affect the loss function the most (Miconi et al., 2018; Zenke et al., 2017; Kirkpatrick et al., 2017; Lee et al., 2017; Serrà et al., 2018). Because these approaches use a single model to encode both global task-general information and local task-specific information, they can over-regularize, preventing the model from learning further tasks. More importantly, Schwarz et al. (2018) found that while these approaches mitigate catastrophic forgetting, they are unable to facilitate knowledge transfer on the benchmark they considered. Ultimately, if a single model must encode both task-generic and task-specific information, it must either saturate or grow in size (Rusu et al., 2016).

In contrast, meta-learning aims to learn the learning process itself (Schmidhuber, 1987; Bengio et al., 1991; Santoro et al., 2016; Ravi & Larochelle, 2016; Andrychowicz et al., 2016; Vinyals et al., 2016; Finn et al., 2017). The literature focuses primarily on few-shot learning, where a task is some variation on a common theme, such as subsets of classes drawn from a shared pool of data (Lake et al., 2015; Vinyals et al., 2016). The meta-learning algorithm adapts a model to a new task given a handful of samples. Recent attention has been devoted to three main approaches. One trains the meta-learner to adapt to a new task by comparing an input to samples from previous tasks (Vinyals et al., 2016; Mishra et al., 2018; Snell et al., 2017). More relevant to our framework are approaches that parameterize the training process through a recurrent neural network that takes the gradient as input and produces a new set of parameters (Ravi & Larochelle, 2016; Santoro et al., 2016; Andrychowicz et al., 2016; Hochreiter et al., 2001). The approach most closely related to us learns an initialization such that the model can adapt to a new task through one or a few gradient updates (Finn et al., 2017; Nichol et al., 2018; Al-Shedivat et al., 2017; Lee & Choi, 2018). In contrast to our work, these methods focus exclusively on few-shot learning, where the gradient path is trivial as only a single or a handful of training steps are allowed, limiting them to settings where the current task is closely related to previous ones.

It is worth noting that the Model Agnostic Meta Learner (MAML: Finn et al., 2017) can be written as $\mathbb{E}_{\tau\sim p(\tau)}\left[f_\tau(\theta_\tau^K)\right]$.[1] As such, it arises as a special case of Leap where only the final parameterization is evaluated in terms of its final performance. Similarly, the Reptile algorithm (Nichol et al., 2018), which proposes to update rule $\theta^0 \leftarrow \theta^0 + \epsilon\left(\mathbb{E}_{\tau\sim p(\tau)}\left[\theta_\tau^K\right] - \theta^0\right)$, can be seen as a naive version of Leap that assumes all task geometries are Euclidean. In particular, Leap reduces to Reptile if $f_\tau$ is removed from the task manifold and the energy metric without stabilizer is used. We find this configuration to perform significantly worse than any other (see section 4.1 and appendix C).

---

[1] MAML differs from Leap in that it evaluates the meta objective on a held-out test set.

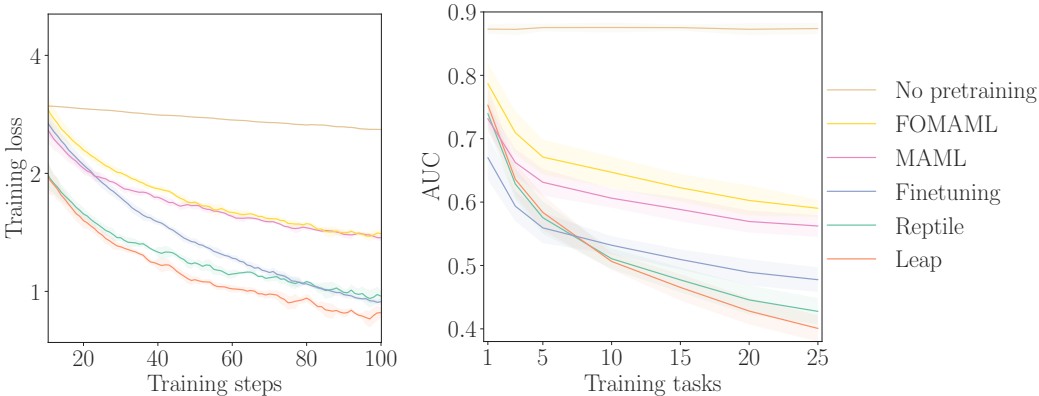

Figure 3: Results on Omniglot. *Left:* Comparison of average learning curves on held-out tasks (across 10 seeds) for 25 tasks in the meta-training set. Curves are moving averages with window size 5. Shading: standard deviation within window. *Right:* AUC across number of tasks in the meta-training set. Shading: standard deviation across 10 seeds.

Related work studying models from a geometric perspective have explored how to interpolate in a generative model's learned latent space (Tosi et al., 2014; Shao et al., 2017; Arvanitidis et al., 2018; Chen et al., 2018; Kumar et al., 2017). Riemann manifolds have also garnered attention in the context of optimization, as a preconditioning matrix can be understood as the instantiation of some Riemann metric (Amari & Nagaoka, 2007; Abbati et al., 2018; Luk & Grosse, 2018).

# 4 EMPIRICAL RESULTS

We consider three experiments with increasingly complex knowledge transfer. We measure transfer learning in terms of final performance and speed of convergence, where the latter is defined as the area under the training error curve. We compare Leap to competing meta-learning methods on the Omniglot dataset by transferring knowledge across alphabets (section 4.1). We study Leap's ability to transfer knowledge over more complex and diverse tasks in a Multi-CV experiment (section 4.2) and finally evaluate Leap on in a demanding reinforcement environment (section 4.3).

## 4.1 OMNIGLOT

The Omniglot (Lake et al., 2015) dataset consists of 50 alphabets, which we define to be distinct tasks. We hold 10 alphabets out for final evaluation and use subsets of the remaining alphabets for meta-learning or pretraining. We vary the number of alphabets used for meta-learning / pretraining from 1 to 25 and compare final performance and rate of convergence on held-out tasks. We compare against no pretraining, multi-headed finetuning, MAML, the first-order approximation of MAML (FOMAML; Finn et al., 2017), and Reptile. We train on a given task for 100 steps, with the exception of MAML where we backpropagate through 5 training steps during meta-training. For Leap, we report performance under the length metric ($d_1$); see appendix C for an ablation study on Leap hyper-parameters. For further details, see appendix D.

Any type of knowledge transfer significantly improves upon a random initialization. MAML exhibits a considerable short-horizon bias (Wu et al., 2018). While FOMAML is trained full trajectories, but because it only leverages gradient information at final iteration, which may be arbitrarily uninformative, it does worse. Multi-headed finetuning is a tough benchmark to beat as tasks are very similar. Nevertheless, for sufficiently rich task distributions, both Reptile and Leap outperform finetuning, with Leap outperforming Reptile as the complexity grows. Notably, the AUC gap between Reptile and Leap grows in the number of training steps (fig. 3), amounting to a 4 percentage point difference in final validation error (table 2). Overall, the relative performance of meta-learners underscores the importance of leveraging geometric information in meta-learning.

Table 1: Results on Multi-CV benchmark. All methods are trained until convergence on held-out tasks. Finetuning is multiheaded. [†] Area under training error curve; scaled to 0–100.[‡]Our implementation. MNIST results omitted; see appendix E, table 4.

| Held-out task | Method | Test (%) | Train (%) | AUC[†] |
|---|---|---|---|---|
| Facescrub | Leap | 19.9 | 0.0 | 11.6 |
| | Finetuning | 32.7 | 0.0 | 13.2 |
| | Progressive Nets[‡] | **18.0** | 0.0 | **8.9** |
| | HAT[‡] | 25.6 | 0.1 | 14.6 |
| | No pretraining | 18.2 | 0.0 | 10.5 |
| Cifar10 | Leap | **21.2** | **10.8** | **17.5** |
| | Finetuning | 27.4 | 13.3 | 20.7 |
| | Progressive Nets[‡] | 24.2 | 15.2 | 24.0 |
| | HAT[‡] | 27.7 | 21.2 | 27.3 |
| | No pretraining | 26.2 | 13.1 | 23.0 |
| SVHN | Leap | **8.4** | **5.6** | **7.5** |
| | Finetuning | 10.9 | 6.1 | 10.5 |
| | Progressive Nets[‡] | 10.1 | 6.3 | 13.8 |
| | HAT[‡] | 10.5 | 5.7 | 8.5 |
| | No pretraining | 10.3 | 6.9 | 11.5 |
| Cifar100 | Leap | **52.0** | **30.5** | **43.4** |
| | Finetuning | 59.2 | 31.5 | 44.1 |
| | Progressive Nets[‡] | 55.7 | 42.1 | 54.6 |
| | HAT[‡] | 62.0 | 49.8 | 58.4 |
| | No pretraining | 54.8 | 33.1 | 50.1 |
| Traffic Signs | Leap | **2.9** | 0.0 | **1.2** |
| | Finetuning | 5.7 | 0.0 | 1.7 |
| | Progressive Nets[‡] | 3.6 | 0.0 | 4.0 |
| | HAT[‡] | 5.4 | 0.0 | 2.3 |
| | No pretraining | 3.6 | 0.0 | 2.4 |

## 4.2 MULTI-CV

Inspired by Serrà et al. (2018), we consider a set of computer vision datasets as distinct tasks. We pretrain on all but one task, which is held out for final evaluation. For details, see appendix E. To reduce the computational burden during meta training, we pretrain on each task in the meta batch for one epoch using the energy metric ($d_2$). We found this to reach equivalent performance to training on longer gradient paths or using the length metric. This indicates that it is sufficient for Leap to see a partial trajectory to correctly infer shared structures across task geometries.

We compare Leap against a random initialization, multi-headed finetuning, a non-sequential version of HAT (Serrà et al., 2018) (i.e. allowing revisits) and a non-sequential version of Progressive Nets (Rusu et al., 2016), where we allow lateral connection between every task. Note that this makes Progressive Nets over 8 times larger in terms of learnable parameters.

The Multi-CV experiment is more challenging both due to greater task diversity and greater complexity among tasks. We report results on held-out tasks in table 1. Leap outperforms all baselines on all but one transfer learning tasks (Facescrub), where Progressive Nets does marginally better than a random initialization owing to its increased parameter count. Notably, while Leap does marginally worse than a random initialization, finetuning and HAT leads to a substantial drop in performance. On all other tasks, Leap converges faster to optimal performance and achieves superior final performance.

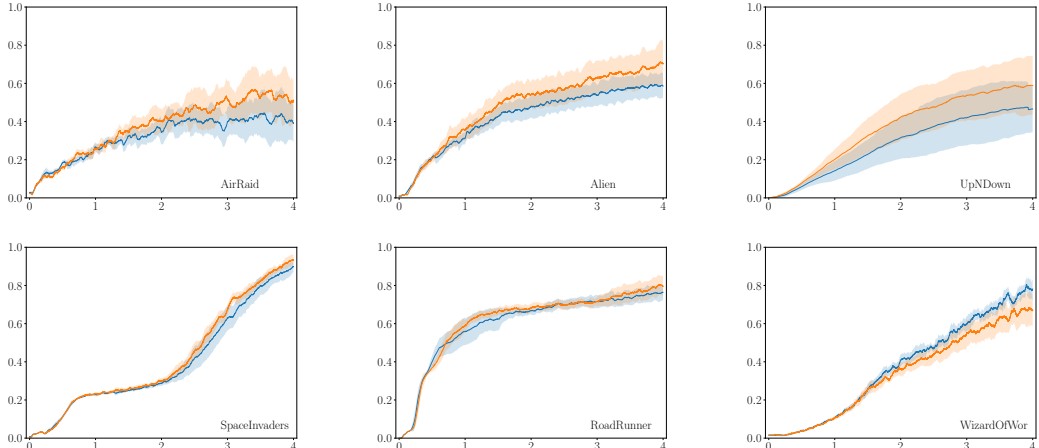

Figure 4: Mean normalized episode scores on Atari games across training steps. Shaded regions depict two standard deviations across ten seeds. Leap (orange) generally outperforms a random initialization (blue), even when the action space is twice as large as during pretraining (table 6, appendix F).

### 4.3 ATARI

To demonstrate that Leap can scale to large problems, both in computational terms and in task complexity, we apply it in a reinforcement learning environment, specifically Atari 2600 games (Bellemare et al., 2013). We use an actor-critic architecture (Sutton et al., 1998) with the policy and the value function sharing a convolutional encoder. We apply Leap with respect to the encoder using the energy metric ($d_2$). During meta training, we sample mini-batches from 27 games that have an action space dimensionality of at most 10, holding out two games with similar action space dimensionality for evaluation, as well as games with larger action spaces (table 6). During meta-training, we train on each task for five million training steps. See appendix F for details.

We train for 100 meta training steps, which is sufficient to see a distinct improvement; we expect a longer meta-training phase to yield further gains. We find that Leap generally outperforms a random initialization. This performance gain is primarily driven by less volatile exploration, as seen by the confidence intervals in fig. 4 (see also fig. 8). Leap finds a useful exploration space faster and more consistently, demonstrating that Leap can find shared structures across a diverse set of complex learning processes. We note that these gains may not cater equally to all tasks. In the case of WizardOfWor (part of the meta-training set), Leap exhibits two modes: in one it performs on par with the baseline, in the other exploration is protracted (fig. 8). This phenomena stems from randomness in the learning process, which renders an observed gradient path relatively less representative. Such randomness can be marginalized by training for longer.

That Leap can outperform a random initialization on the pretraining set (AirRaid, UpNDown) is perhaps not surprising. More striking is that it exhibits the same behavior on out-of-distribution tasks. In particular, Alien, Gravitar and RoadRunner all have at least 50% larger state space than anything encountered during pretraining (appendix F, table 6), yet Leap outperforms a random initialization. This suggests that transferring knowledge at a higher level of abstraction, such as in the space of gradient paths, generalizes to unseen task variations as long as underlying learning dynamics agree.

## 5 CONCLUSIONS

Transfer learning typically ignores the learning process itself, restricting knowledge transfer to scenarios where target tasks are very similar to source tasks. In this paper, we present Leap, a framework for knowledge transfer at a higher level of abstraction. By formalizing knowledge transfer as minimizing the expected length of gradient paths, we propose a method for meta-learning that scales to highly demanding problems. We find empirically that Leap has superior generalizing properties to finetuning and competing meta-learners.

ACKNOWLEDGMENTS

The authors would like to thank anonymous reviewers for their comments. This work was supported by The Alan Turing Institute under the EPSRC grant EP/N510129/1.

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

APPENDIX

## A   PROOF OF THEOREM 1

*Proof.* We first establish that, for all $s$,

$$\mathbb{E}_\tau \, d(\theta^0_{s+1}, M_\tau) = F(\theta^0_{s+1}) = \bar{F}(\theta^0_{s+1}; \Psi_{s+1}) \leq \bar{F}(\theta^0_s; \Psi_s) = F(\theta^0_s) = \mathbb{E}_\tau \, d(\theta^0_s, M_\tau),$$

with strict inequality for at least some $s$. Because $\{\beta_s\}_{s=1}^\infty$ satisfies the gradient descent criteria, it follows that the sequence $\{\theta^0_s\}_{s=1}^\infty$ is convergent. To complete the proof we must show that this limit point lies in $\Theta$. To this end, we show that for $\beta_s$ sufficiently small, for all $s$, $\lim_{i\to\infty} \theta^i_{s+1} = \lim_{i\to\infty} \theta^i_s$. That is, each updated initialization incrementally reduces the expected gradient path length while converging to the same limit point as $\theta^0_0$. Since $\theta^0_0 \in \Theta$ by assumption, we obtain $\theta^0_s \in \Theta$ for all $s$ as an immediate consequence.

To establish $\mathbb{E}_\tau \, d(\theta^0_{s+1}, M_\tau) \leq \mathbb{E}_\tau \, d(\theta^0_s, M_\tau)$, with strict inequality for some $s$, let

$$
\begin{aligned}
z^i_\tau &= (\theta^{s,i}_\tau, f_\tau(\theta^{s,i}_\tau)) & x^i_\tau &= (\theta^{s+1,i}_\tau, f_\tau(\theta^{s+1,i}_\tau)) \\
h^i_\tau &= (\psi^{s,i+1}_\tau, f_\tau(\psi^{s,i+1}_\tau)) & y^i_\tau &= (\psi^{s+1,i+1}_\tau, f_\tau(\psi^{s+1,i+1}_\tau)),
\end{aligned}
$$

with $\psi^{s,i+1}_\tau = \theta^{s,i+1}_\tau$. Denote by $\mathbb{E}_{\tau,i}$ the expectation over gradient paths, $\mathbb{E}_{\tau\sim p(\tau)}\sum_{i=1}^{K_\tau}$. Note that

$$
\begin{aligned}
\bar{F}(\theta^0_s, \Psi_s) &= \mathbb{E}_{\tau,i} \, \|h^i_\tau - z^i_\tau\|^p_2 & \bar{F}(\theta^0_s, \Psi_{s+1}) &= \mathbb{E}_{\tau,i} \, \|y^i_\tau - z^i_\tau\|^p_2 \\
\bar{F}(\theta^0_{s+1}, \Psi_s) &= \mathbb{E}_{\tau,i} \, \|h^i_\tau - x^i_\tau\|^p_2 & \bar{F}(\theta^0_{s+1}, \Psi_{s+1}) &= \mathbb{E}_{\tau,i} \, \|y^i_\tau - x^i_\tau\|^p_2
\end{aligned}
$$

with $p = 2$ defining the meta objective in terms of the gradient path energy and $p = 1$ in terms of the gradient path length. As we are exclusively concerned with the Euclidean norm, we omit the subscript. By assumption, every $\beta_s$ is sufficiently small to satisfy the gradient descent criteria $\bar{F}(\theta^0_s; \Psi_s) \geq \bar{F}(\theta^0_{s+1}; \Psi_s)$. Adding and subtracting $\bar{F}(\theta^0_{s+1}, \Psi_{s+1})$ to the RHS, we have

$$
\begin{aligned}
\mathbb{E}_{\tau,i} \, \|h^i_\tau - z^i_\tau\|^p \geq \mathbb{E}_{\tau,i} \, \|h^i_\tau - x^i_\tau\|^p \\
= \mathbb{E}_{\tau,i} \, \|y^i_\tau - x^i_\tau\|^p + \|h^i_\tau - x^i_\tau\|^p - \|y^i_\tau - x^i_\tau\|^p.
\end{aligned}
$$

It follows that $\mathbb{E}_\tau \, d(\theta^0_s, M_\tau) \geq \mathbb{E}_\tau \, d(\theta^0_{s+1}, M_\tau)$ if $\mathbb{E}_{\tau,i} \, \|h^i_\tau - x^i_\tau\|^p \geq \mathbb{E}_{\tau,i} \, \|y^i_\tau - x^i_\tau\|^p$. As our main concern is existence, we will show something stronger, namely that there exists $\alpha^i_\tau$ such that

$$\|h^i_\tau - x^i_\tau\|^p \geq \|y^i_\tau - x^i_\tau\|^p \qquad \forall i, \tau, s, p$$

with at least one such inequality strict for some $i, \tau, s$, in which case $d_p(\theta^0_{s+1}, M_\tau) < d_p(\theta^0_s, M_\tau)$ for any $p \in \{1, 2\}$. We proceed by establishing the inequality for $p = 2$ and obtain $p = 1$ as an immediate consequence of monotonicity of the square root. Expanding $\|h^i_\tau - x^i_\tau\|^2$ we have

$$
\begin{aligned}
\|h^i_\tau - x^i_\tau\|^2 - \|y^i_\tau - x^i_\tau\|^2 &= \|(h^i_\tau - z^i_\tau) + (z^i_\tau - x^i_\tau)\|^2 - \|y^i_\tau - x^i_\tau\|^2 \\
&= \|h^i_\tau - z^i_\tau\|^2 + 2\langle h^i_\tau - z^i_\tau, z^i_\tau - x^i_\tau \rangle + \|z^i_\tau - x^i_\tau\|^2 - \|y^i_\tau - x^i_\tau\|^2.
\end{aligned}
$$

Every term except $\|z^i_\tau - x^i_\tau\|^2$ can be minimized by choosing $\alpha^i_\tau$ small, whereas $\|z^i_\tau - x^i_\tau\|^2$ is controlled by $\beta_s$. Thus, our strategy is to make all terms except $\|z^i_\tau - x^i_\tau\|^2$ small, for a given $\beta_s$, by placing an upper bound on $\alpha^i_\tau$. We first show that $\|h^i_\tau - z^i_\tau\|^2 - \|y^i_\tau - x^i_\tau\|^2 = O\left({\alpha^i_\tau}^2\right)$. Some care is needed as the $(n + 1)$th dimension is the loss value associated with the other $n$ dimensions. Define $\hat{z}^i_\tau = \theta^{s,i}_\tau$, so that $z^i_\tau = (\hat{z}^i_\tau, f_\tau(\hat{z}^i_\tau))$. Similarly define $\hat{x}^i_\tau, \hat{h}^i_\tau$, and $\hat{y}^i_\tau$ to obtain

$$\|h_\tau^i - z_\tau^i\|^2 = \|\hat{h}_\tau^i - \hat{z}_\tau^i\|^2 + (f_\tau(\hat{h}_\tau^i) - f_\tau(\hat{z}_\tau^i))^2$$
$$\|y_\tau^i - x_\tau^i\|^2 = \|\hat{y}_\tau^i - \hat{x}_\tau^i\|^2 + (f_\tau(\hat{y}_\tau^i) - f_\tau(\hat{x}_\tau^i))^2$$
$$2\langle h_\tau^i - z_\tau^i, z_\tau^i - x_\tau^i \rangle = 2\langle \hat{h}_\tau^i - \hat{z}_\tau^i, \hat{z}_\tau^i - \hat{x}_\tau^i \rangle + (f_\tau(\hat{h}_\tau^i) - f_\tau(\hat{z}_\tau^i))(f_\tau(\hat{z}_\tau^i) - f_\tau(\hat{x}_\tau^i)).$$

Consider $\|\hat{h}_\tau^i - \hat{z}_\tau^i\|^2 - \|\hat{y}_\tau^i - \hat{x}_\tau^i\|^2$. Note that $\hat{h}_\tau^i = \hat{z}_\tau^i - \alpha_\tau^i g(\hat{z}_\tau^i)$, where $g(\hat{z}_\tau^i) = S_\tau^{s,i} \nabla f(\hat{z}_\tau^i)$, and similarly $\hat{y}_\tau^i = \hat{x}_\tau^i - \alpha_\tau^i g(\hat{x}_\tau^i)$ with $g(\hat{x}_\tau^i) = S_\tau^{s+1,i} \nabla f(\hat{x}_\tau^i)$. Thus, $\|\hat{h}_\tau^i - \hat{z}_\tau^i\|^2 = {\alpha_\tau^i}^2 \|g(\hat{z}_\tau^i)\|^2$ and similarly for $\|\hat{y}_\tau^i - \hat{x}_\tau^i\|^2$, so

$$\|\hat{h}_\tau^i - \hat{z}_\tau^i\|^2 - \|\hat{y}_\tau^i - \hat{x}_\tau^i\|^2 = \left(\alpha_\tau^i\right)^2 \left( \|g(\hat{z}_\tau^i)\|^2 - \|g(\hat{x}_\tau^i)\|^2 \right) = O\left( \left(\alpha_\tau^i\right)^2 \right).$$

Now consider $(f_\tau(\hat{h}_\tau^i) - f_\tau(\hat{z}_\tau^i))^2 - (f_\tau(\hat{y}_\tau^i) - f_\tau(\hat{x}_\tau^i))^2$. Using the above identities and first-order Taylor series expansion, we have

$$(f_\tau(\hat{h}_\tau^i) - f_\tau(\hat{z}_\tau^i))^2 = \left( \nabla f_\tau(\hat{z}_\tau^i)^T (\hat{h}_\tau^i - \hat{z}_\tau^i) + O\left(\alpha_\tau^i\right) \right)^2$$
$$= \left( -\alpha_\tau^i \nabla f_\tau(\hat{z}_\tau^i)^T g(\hat{z}_\tau^i) + O\left(\alpha_\tau^i\right) \right)^2 = O\left( \left(\alpha_\tau^i\right)^2 \right),$$

and similarly for $(f_\tau(\hat{y}_\tau^i) - f_\tau(\hat{x}_\tau^i))^2$. As such, $\|h_\tau^i - z_\tau^i\|^2 - \|y_\tau^i - x_\tau^i\|^2 = O\left( \left(\alpha_\tau^i\right)^2 \right)$.

Finally, consider the inner product $\langle h_\tau^i - z_\tau^i, z_\tau^i - x_\tau^i \rangle$. From above we have that $(f_\tau(\hat{h}_\tau^i) - f_\tau(\hat{z}_\tau^i))^2 = -\alpha_\tau^i(\nabla f_\tau(\hat{z}_\tau^i)^T g(\hat{z}_\tau^i) - R_\tau^i) = -\alpha_\tau^i \xi_\tau^i$, where $R_\tau^i$ denotes an upper bound on the residual. We extend $g$ to operate on $z_\tau^i$ by defining $\tilde{g}(z_\tau^i) = (g(\hat{z}_\tau^i), \xi_\tau^i)$. Returning to $\|h_\tau^i - x_\tau^i\|^2 - \|y_\tau^i - x_\tau^i\|^2$, we have

$$\|h_\tau^i - x_\tau^i\|^2 - \|y_\tau^i - x_\tau^i\|^2 = \|z_\tau^i - x_\tau^i\|^2 + 2\langle h_\tau^i - z_\tau^i, z_\tau^i - x_\tau^i \rangle + O\left( \left(\alpha_\tau^i\right)^2 \right)$$
$$= \|z_\tau^i - x_\tau^i\|^2 - 2\alpha_\tau^i \langle \tilde{g}(z_\tau^i), z_\tau^i - x_\tau^i \rangle + O\left( \left(\alpha_\tau^i\right)^2 \right).$$

The first term is non-negative, and importantly, always non-zero whenever $\beta_s \neq 0$. Furthermore, $\alpha_\tau^i$ can always be made sufficiently small for $\|z_\tau^i - x_\tau^i\|^2$ to dominate the residual, so we can focus on the inner product $\langle \tilde{g}(z_\tau^i), z_\tau^i - x_\tau^i \rangle$. If it is negative, all terms are positive and we have $\|h_\tau^i - x_\tau^i\|^2 \geq \|y_\tau^i - x_\tau^i\|^2$ as desired. If not, $\|z_\tau^i - x_\tau^i\|^2$ dominates if

$$\alpha_\tau^i \leq \frac{\|z_\tau^i - x_\tau^i\|^2}{2\langle \tilde{g}(z_\tau^i), z_\tau^i - x_\tau^i \rangle} \in (0, \infty).$$

Thus, for $\alpha_\tau^i$ sufficiently small, we have $\|h_\tau^i - x_\tau^i\|^2 \geq \|y_\tau^i - x_\tau^i\|^2 \ \forall i, \tau, s$, with strict inequality whenever $\langle \tilde{g}(z_\tau^i), z_\tau^i - x_\tau^i \rangle < 0$ or the bound on $\alpha_\tau^i$ holds strictly. This establishes $d_2(\theta_{s+1}^0, M_\tau) \leq d_2(\theta_s^0, M_\tau)$ for all $\tau, s$, with strict inequality for at least some $\tau, s$. To also establish it for the gradient path length ($p = 1$), taking square roots on both sides of $\|h_\tau^i - x_\tau^i\|^2 \geq \|y_\tau^i - x_\tau^i\|^2$ yields the desired results, and so $\|h_\tau^i - x_\tau^i\|^p \geq \|y_\tau^i - x_\tau^i\|^p$ for $p \in \{1, 2\}$, and therefore

$$d(\theta_{s+1}^0, M_\tau) = \bar{F}(\theta_{s+1}^0; \Psi_{s+1}) \leq \bar{F}(\theta_s^0; \Psi_s) = d(\theta_s^0, M_\tau) \quad \forall \tau, s$$

with strict inequality for at least some $\tau, s$, in particular whenever $\beta_s \neq 0$ and $\alpha_\tau^i$ sufficiently small.

Then, to see that the limit point of $\Psi_{s+1}$ is the same as that of $\Psi_s$ for $\beta_s$ sufficiently small, note that $x_\tau^i = y_\tau^{i-1}$. As before, by the gradient descent criteria, $\beta_s$ is such that

$$\mathbb{E}_{\tau,i} \|h_\tau^i - x_\tau^i\|^p = \mathbb{E}_{\tau,i} \|h_\tau^i - y_\tau^{i-1}\|^p \leq \mathbb{E}_{\tau,i} \|h_\tau^i - z_\tau^i\|^p = \mathbb{E}_{\tau,i} \left(\alpha_\tau^i\right)^p \|\tilde{g}(z_\tau^i)\|^p.$$

Define $\epsilon_\tau^i$ as the noise residual from the expectation; each $y_\tau^{i-1}$ is bounded by $\|h_\tau^i - y_\tau^{i-1}\|^p \le \left(\alpha_\tau^i\right)^p \|\tilde{g}(z_\tau^i)\|^p + \epsilon_\tau^i$. For $\beta_s$ small this noise component vanishes, and since $\{\alpha_\tau^i\}_i$ is a converging sequence, the bound on $y_\tau^{i-1}$ grows increasingly tight. It follows then that $\{\theta_{s+1}^i\}_{i=1}^\infty$ converges to the same limit point as $\{\theta_s^i\}_{i=1}^\infty$, yielding $\theta_{s+1}^0 \in \Theta$ for all $s$, as desired. ∎

# B ABLATION STUDY: APPROXIMATING JACOBIANS $J^i(\theta^0)$

To understand the role of the Jacobians, note that (we drop task subscripts for simplicity)

$$J^{i+1}(\theta^0) = \left(I_n - \alpha^i S^i H_f(\theta^i)\right) J^i(\theta^0) = \prod_{j=0}^{i} \left(I_n - \alpha^j S^j H_f(\theta^j)\right)$$

$$= I_n - \sum_{j=0}^{i} \alpha^i S^i H_f(\theta^i) + O\left(\left(\alpha^i\right)^2\right),$$

where $H_f(\theta^j)$ denotes the Hessian of $f$ at $\theta^j$. Thus, changes to $\theta^{i+1}$ are translated into $\theta^0$ via all intermediary Hessians. This makes the Jacobians memoryless up to second-order curvature. Importantly, the effect of curvature can directly be controlled via $\alpha^i$, and by choosing $\alpha^i$ small we can ensure $J^i(\theta^0) \approx I_n$ to be a arbitrary precision. In practice, this approximation works well (c.f. Finn et al., 2017; Nichol et al., 2018). Moreover, as a practical matter, if the alternative is some other approximation to the Hessians, the amount of noise injected grows exponentially with every iteration. The problem of devising an accurate low-variance estimator for the $J^i(\theta^0)$ is highly challenging and beyond the scope of this paper.

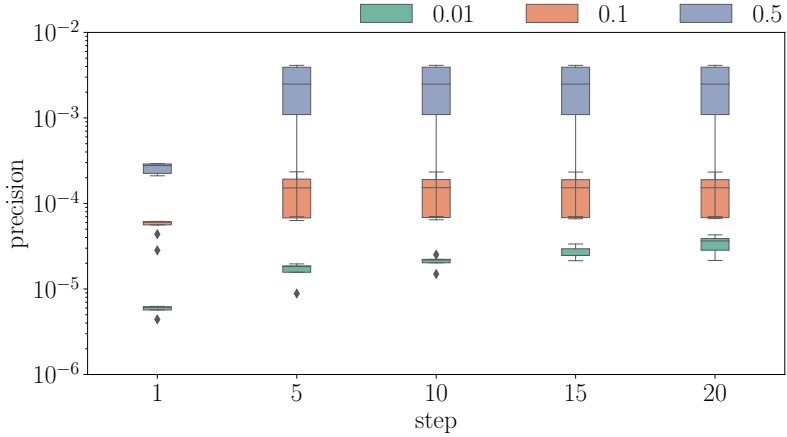

Figure 5: Relative precision of Jacobian approximation. Precision is calculated for the Jacobian of the first layer, across different learning rates (colors) and gradient steps.

To understand how this approximation limits our choice of learning rates $\alpha^i$, we conduct an ablation study in the Omniglot experiment setting. We are interested in the relative precision of the identity approximation under different learning rates and across time steps, which we define as

$$\rho\left(i, \{\alpha^j\}_{j=0}^i\right) = \frac{\|I_n - J^i(\theta^0)\|_1}{\|J^i(\theta^0)\|_1},$$

where the norm is the Schatten 1-norm. We use the same four-layer convolutional neural network as in the Omniglot experiment (appendix D). For each choice of learning rate, we train a model from a random initialization for 20 steps and compute $\rho$ every 5 steps. Due to exponential growth of memory

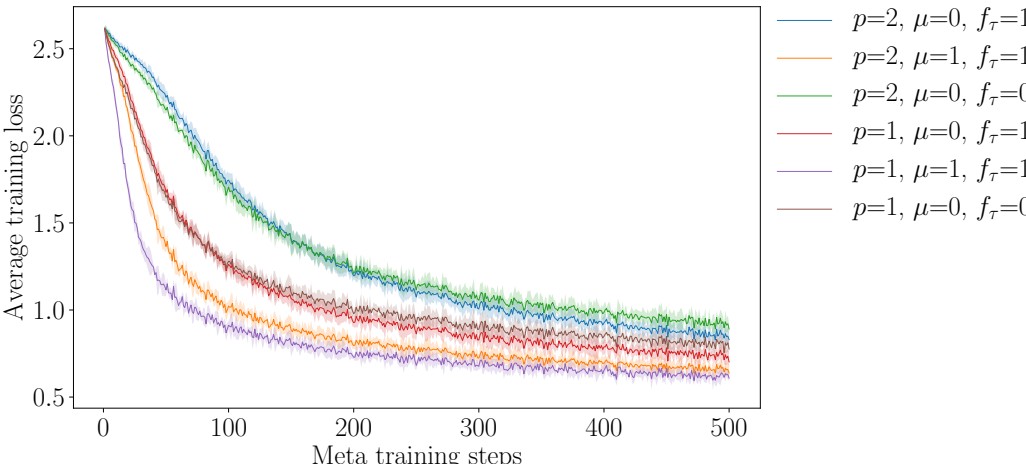

Figure 6: Average task training loss over meta-training steps. $p$ denotes the $\bar{d}_p$ used in the meta objective, $\mu = 1$ the use of the stabilizer, and $f_\tau = 1$ the inclusion of the loss in the task manifold.

consumption, we were unable to compute $\rho$ for more than 20 gradient steps. We report the relative precision of the first convolutional layer. We do not report the Jacobian with respect to other layers, all being considerably larger, as computing their Jacobians was too costly. We computed $\rho$ for all layers on the first five gradient steps and found no significant variation in precision across layers. Consequently, we prioritize reporting how precision varies with the number of gradient steps. As in the main experiments, we use stochastic gradient descent. We evaluate $\alpha^i = \alpha \in \{0.01, 0.1, 0.5\}$ across 5 different tasks. Figure 5 summarizes our results.

Reassuringly, we find the identity approximation to be accurate to at least the fourth decimal for learning rates we use in practice, and to the third decimal for the largest learning rate (0.5) we were able to converge with. Importantly, except for the smallest learning rate, the quality of the approximation is constant in the number of gradient steps. The smallest learning rate that exhibits some deterioration on the fifth decimal, however larger learning rates provide an upper bound that is constant on the fourth decimal, indicating that this is of minor concern. Finally, we note that while these results suggest the identity approximation to be a reasonable approach on the class of problems we consider, other settings may put stricter limits on the effective size of learning rates.

## C   ABLATION STUDY: LEAP HYPER-PARAMETERS

As Leap is a general framework, we have several degrees of freedom in specifying a meta learner. In particular, we are free to choose the task manifold structure, the gradient path distance metric, $d_p$, and whether to incorporate stabilizers. These are non-trivial choices and to ascertain the importance of each, we conduct an ablation study. We vary (a) the task manifold between using the full loss surface and only parameter space, (b) the gradient path distance metric between using the energy or length, and (c) inclusion of the stabilizer $\mu$ in the meta objective. We stay as close as possible to the set-up used in the Omniglot experiment (appendix D), fixing the number of pretraining tasks to 20 and perform 500 meta gradient updates. All other hyper-parameters are the same.

Our ablation study indicates that the richer the task manifold and the more accurate the gradient path length is approximated, the better Leap performs (fig. 6). Further, adding a stabilizer has the intended effect and leads to significantly faster convergence. The simplest configuration, defined in terms of the gradient path energy and with the task manifold identifies as parameter space, yields a meta gradient equivalent to the update rule used in Reptile. We find this configuration to be less efficient in terms of convergence and we observe a significant deterioration in performance. Extending the task manifold to the loss surface does not improve meta-training convergence speed, but does cut prediction error in half. Adding the stabilizer significantly speeds up convergence. These conclusions also hold under the gradient path length as distance measure, and in general using the gradient path length does better than using the gradient path energy as the distance measure.

## D  EXPERIMENT DETAILS: OMNIGLOT

Table 2: Mean test error after 100 training steps on held out evaluation tasks.[†]Multi-headed finetuning.

| Method
No. Pretraining tasks | Leap | Reptile | Finetuning[†] | MAML | FOMAML | No pretraining |
|---|---|---|---|---|---|---|
| 1 | 62.3 | 59.8 | 46.5 | 64.0 | 64.5 | 82.3 |
| 3 | 46.5 | 46.5 | 36.0 | 56.2 | 59.0 | 82.3 |
| 5 | 40.3 | 41.4 | 32.5 | 50.1 | 53.0 | 82.5 |
| 10 | 32.6 | 35.6 | 28.7 | 49.3 | 49.6 | 82.9 |
| 15 | 29.6 | 33.3 | 26.9 | 45.5 | 47.8 | 82.6 |
| 20 | 26.0 | 30.8 | 24.7 | 41.7 | 45.4 | 82.6 |
| 25 | 24.8 | 29.4 | 23.5 | 42.9 | 44.0 | 82.8 |

Table 3: Summary of hyper-parameters for Omniglot. "Meta" refers to the outer training loop, "task" refers to the inner training loop.

| | Leap | Finetuning | Reptile | MAML | FOMAML | No pretraining |
|---|---|---|---|---|---|---|
| **Meta training** | | | | | | |
| Learning rate | 0.1 | — | 0.1 | 0.5 | 0.5 | — |
| Training steps | 1000 | 1000 | 1000 | 1000 | 1000 | — |
| Batch size (tasks) | 20 | 20 | 20 | 20 | 20 | — |
| **Task training** | | | | | | |
| Learning rate | 0.1 | 0.1 | 0.1 | 0.1 | 0.1 | — |
| Training steps | 100 | 100 | 100 | 5 | 100 | — |
| Batch size (samples) | 20 | 20 | 20 | 20 | 20 | — |
| **Task evaluation** | | | | | | |
| Learning rate | 0.1 | 0.1 | 0.1 | 0.1 | 0.1 | 0.1 |
| Training steps | 100 | 100 | 100 | 100 | 100 | 100 |
| Batch size (samples) | 20 | 20 | 20 | 20 | 20 | 20 |

Omniglot contains 50 alphabets, each with a set of characters that in turn have 20 unique samples. We treat each alphabet as a distinct task and pretrain on up to 25 alphabets, holding out 10 out for final evaluation. We use data augmentation on all tasks to render the problem challenging. In particular, we augment any image with a random affine transformation by (a) random sampling a scaling factor between $[0.8, 1.2]$, (b) random rotation between $[0, 360)$, and (c) randomly cropping the height and width by a factor between $[-0.2, 0.2]$ in each dimension. This setup differs significantly from previous protocols (Vinyals et al., 2016; Finn et al., 2017), where tasks are defined by selecting different permutations of characters and restricting the number of samples available for each character.

We use the same convolutional neural network architecture as in previous works (Vinyals et al., 2016; Schwarz et al., 2018). This model stacks a module, comprised of a $3 \times 3$ convolution with 64 filters, followed by batch-normalization, ReLU activation and $2 \times 2$ max-pooling, four times. All images are downsampled to $28 \times 28$, resulting in a $1 \times 1 \times 64$ feature map that is passed on to a final linear layer. We define a task as a 20-class classification problem with classes drawn from a distinct alphabet. For alphabets with more than 20 characters, we pick 20 characters at random, alphabets with fewer characters (4) are dropped from the task set. On each task, we train a model using stochastic gradient descent. For each model, we evaluated learning rates in the range $[0.001, 0.01, 0.1, 0.5]$; we found 0.1 to be the best choice in all cases. See table 3 for further hyper-parameters.

Table 4: Transfer learning results on Multi-CV benchmark. All methods are trained until convergence on held-out tasks. [†]Area under training error curve; scaled to 0–100. [‡]Our implementation.

| Held-out task | Method | Test (%) | Train (%) | AUC[†] |
|---|---|---|---|---|
| Facescrub | Leap | 19.9 | 0.0 | 11.6 |
| | Finetuning | 32.7 | 0.0 | 13.2 |
| | Progressive Nets[‡] | **18.0** | 0.0 | **8.9** |
| | HAT[‡] | 25.6 | 0.1 | 14.6 |
| | No pretraining | 18.2 | 0.0 | 10.5 |
| NotMNIST | Leap | **5.3** | **0.6** | **2.9** |
| | Finetuning | 5.4 | 2.0 | 4.4 |
| | Progressive Nets[‡] | 5.4 | 3.1 | 3.7 |
| | HAT[‡] | 6.0 | 2.8 | 5.4 |
| | No pretraining | 5.4 | 2.6 | 5.1 |
| MNIST | Leap | **0.7** | 0.1 | **0.6** |
| | Finetuning | 0.9 | 0.1 | 0.8 |
| | Progressive Nets[‡] | 0.8 | **0.0** | 0.7 |
| | HAT[‡] | 0.8 | 0.3 | 1.2 |
| | No pretraining | 0.9 | 0.2 | 1.0 |
| Fashion MNIST | Leap | **8.0** | 4.2 | **6.8** |
| | Finetuning | 8.9 | **3.8** | 7.0 |
| | Progressive Nets[‡] | 8.7 | 5.4 | 9.2 |
| | HAT[‡] | 9.5 | 5.5 | 8.1 |
| | No pretraining | 8.4 | 4.7 | 7.8 |
| Cifar10 | Leap | **21.2** | **10.8** | **17.5** |
| | Finetuning | 27.4 | 13.3 | 20.7 |
| | Progressive Nets[‡] | 24.2 | 15.2 | 24.0 |
| | HAT[‡] | 27.7 | 21.2 | 27.3 |
| | No pretraining | 26.2 | 13.1 | 23.0 |
| SVHN | Leap | **8.4** | **5.6** | **7.5** |
| | Finetuning | 10.9 | 6.1 | 10.5 |
| | Progressive Nets[‡] | 10.1 | 6.3 | 13.8 |
| | HAT[‡] | 10.5 | 5.7 | 8.5 |
| | No pretraining | 10.3 | 6.9 | 11.5 |
| Cifar100 | Leap | **52.0** | **30.5** | **43.4** |
| | Finetuning | 59.2 | 31.5 | 44.1 |
| | Progressive Nets[‡] | 55.7 | 42.1 | 54.6 |
| | HAT[‡] | 62.0 | 49.8 | 58.4 |
| | No pretraining | 54.8 | 33.1 | 50.1 |
| Traffic Signs | Leap | **2.9** | 0.0 | **1.2** |
| | Finetuning | 5.7 | 0.0 | 1.7 |
| | Progressive Nets[‡] | 3.6 | 0.0 | 4.0 |
| | HAT[‡] | 5.4 | 0.0 | 2.3 |
| | No pretraining | 3.6 | 0.0 | 2.4 |

We meta-train for 1000 steps unless otherwise noted; on each task we train for 100 steps. Increasing the number of steps used for task training yields similar results, albeit at greater computational expense. For each character in an alphabet, we hold out 5 samples in order to create a task validation set.

Table 5: Summary of hyper-parameters for Multi-CV. "Meta" refers to the outer training loop, 'task' refers to the inner training loop.

|  | Leap | Finetuning | Progressive Nets | HAT | No pretraining |
|---|---|---|---|---|---|
| **Meta training** | | | | | |
| Learning rate | 0.01 | — | — | — | — |
| Training steps | 1000 | 1000 | 1000 | 1000 | — |
| Batch size | 10 | 10 | 10 | 10 | — |
| **Task training** | | | | | |
| Learning rate | 0.1 | 0.1 | 0.1 | 0.1 | — |
| Max epochs | 1 | 1 | 1 | 1 | — |
| Batch size | 32 | 32 | 32 | 32 | — |
| **Task evaluation** | | | | | |
| Learning rate | 0.1 | 0.1 | 0.1 | 0.1 | 0.1 |
| Training epochs | 100 | 100 | 100 | 100 | 100 |
| Batch size | 32 | 32 | 32 | 32 | 32 |

## E  EXPERIMENT DETAILS: MULTI-CV

We allow different architectures between tasks by using different final linear layers for each task. We use the same convolutional encoder as in the Omniglot experiment (appendix D). Leap learns an initialization for the convolutional encoder; on each task, the final linear layer is always randomly initialized. We compare Leap against (a) a baseline with no pretraining, (b) multitask finetuning, (c) HAT (Serrà et al., 2018), and (d) Progressive Nets (Rusu et al., 2016). For HAT, we use the original formulation, but allow multiple task revisits (until convergence). For Progressive Nets, we allow lateral connections between all tasks and multiple task revisits (until convergence). Note that this makes Progressive Nets over 8 times larger in terms of learnable parameters than the other models. inproceedings We train using stochastic gradient descent with cosine annealing (Loshchilov & Hutter, 2017). During meta training, we sample a batch of 10 tasks at random from the pretraining set and train until the early stopping criterion is triggered or the maximum amount of epochs is reached (see table 5). We used the same interval for selecting learning rates as in the Omniglot experiment (appendix D). Only Leap benefited from using more than 1 epoch as the upper limit on task training steps during pretraining. In the case of Leap, the initialization is updated after all tasks in the meta batch has been trained to convergence; for other models, there is no distinction between initialization and task parameters. On a given task, training is stopped if the maximum number of epochs is reached (table 5) or if the validation error fails to improve over 10 consecutive gradient steps. Similarly, meta training is stopped once the mean validation error fails to improve over 10 consecutive meta training batches. We use Adam (Kingma & Ba, 2015) for the meta gradient update with a constant learning rate of 0.01. We use no dataset augmentation. MNIST images are zero padded to have $32 \times 32$ images; we use the same normalizations as Serrà et al. (2018).

## F  EXPERIMENT DETAILS: ATARI

We use the same network as in Mnih et al. (2013), adopting it to actor-critic algorithms by estimating both value function and policy through linear layers connected to the final output of a shared convolutional network. Following standard practice, we use downsampled $84 \times 84 \times 3$ RGB images as input. Leap is applied with respect to the convolutional encoder (as final linear layers vary in size across environments). We use all environments with an action space of at most 10 as our pretraining pool, holding out Breakout and SpaceInvaders. During meta training, we sample a batch of 16 games at random from a pretraining pool of 27 games. On each game in the batch, a network is initialized using the shared initialization and trained independently for 5 million steps, accumulating the meta gradient across games on the fly. Consequently, in any given episode, the baseline and Leap differs

only with respect to the initialization of the convolutional encoder. We trained Leap for 100 steps, equivalent to training 1600 agents for 5 million steps. The meta learned initialization was evaluated on the held-out games, a random selection of games seen during pretraining, and a random selection of games with action spaces larger than 10 (table 6). On each task, we use a batch size of 32, an unroll length of 5 and update the model parameters with RMSProp (using $\epsilon = 10^{-4}, \alpha = 0.99$) with a learning rate of $10^{-4}$. We set the entropy cost to 0.01 and clip the absolute value of the rewards to maximum 5.0. We use a discounting factor of 0.99.

Table 6: Evaluation environment characteristics. [†]Calculated on baseline (no pretraining) data.

| Environment | Action Space | Mean Reward[†] | Standard Deviation[†] | Pretraining Env |
|---|---|---|---|---|
| AirRaid | 6 | 2538 | 624 | Y |
| UpNDown | 6 | 52417 | 2797 | Y |
| WizardOfWor | 10 | 2531 | 182 | Y |
| Breakout | 4 | 338 | 13 | N |
| SpaceInvaders | 6 | 1065 | 103 | N |
| Asteroids | 14 | 1760 | 139 | N |
| Alien | 18 | 1280 | 182 | N |
| Gravitar | 18 | 329 | 15 | N |
| RoadRunner | 18 | 29593 | 2890 | N |

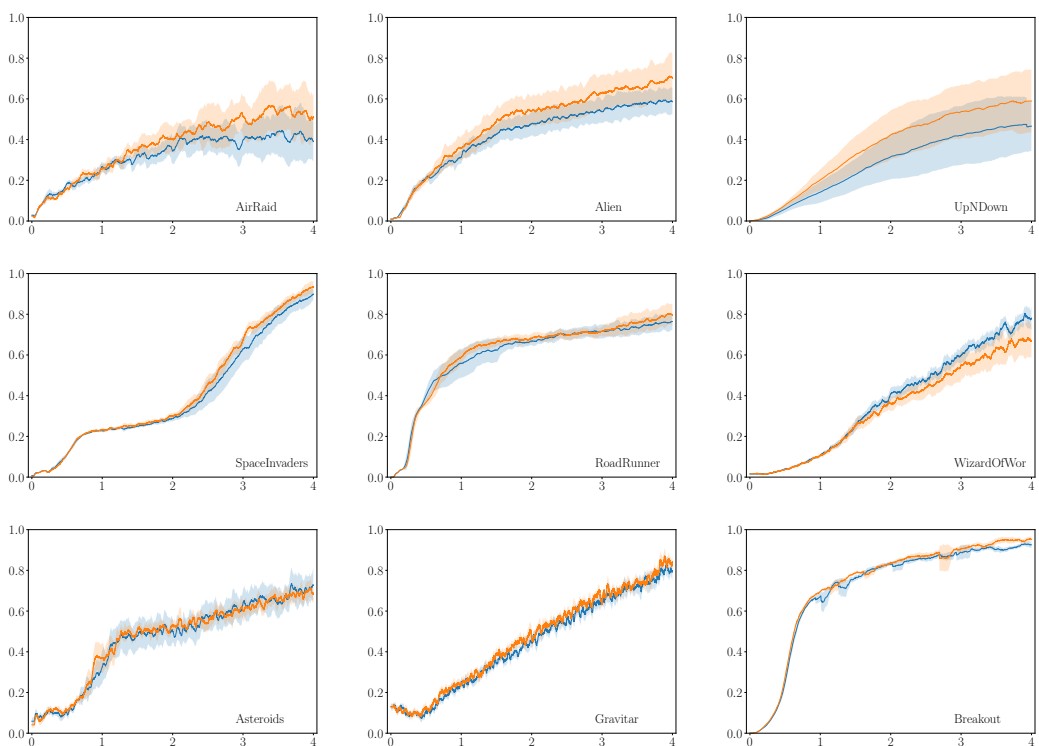

Figure 7: Mean normalized episode scores on Atari games across training steps. Scores are reported as moving average over 500 episodes. Shaded regions depict two standard deviations across ten seeds. KungFuMaster, RoadRunner and Krull have action state spaces that are twice as large as the largest action state encountered during pretraining. Leap (orange) generally outperforms a random initialization, except for WizardOfWor, where a random initialization does better on average due to outlying runs under Leap's initialization.

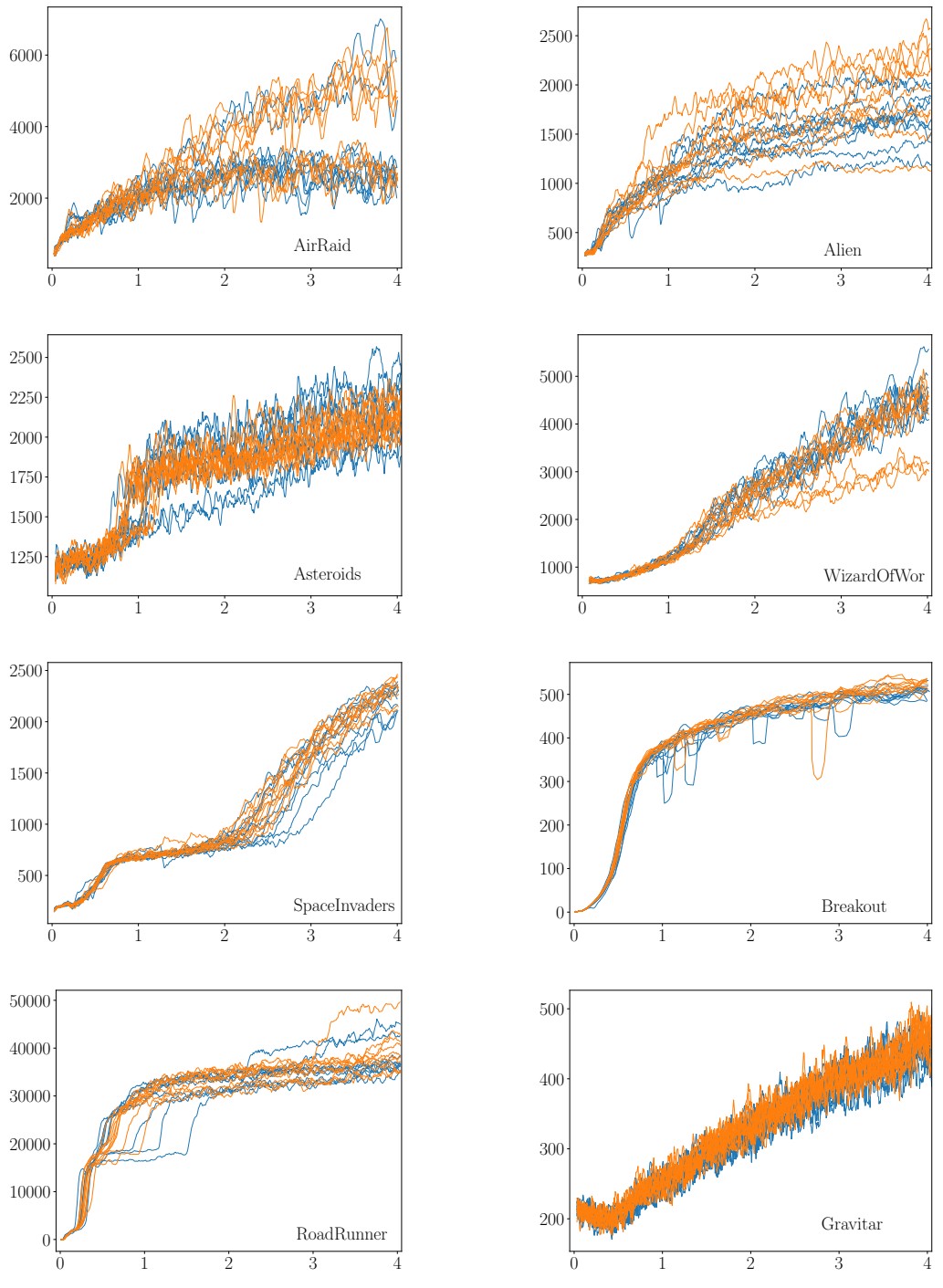

Figure 8: Mean episode scores on Atari games across training steps for different runs. Scores are reported as moving average over 500 episodes. Leap (orange) outperforms a random initialization by being less volatile.

