# OpenReview forum: "Transferring Knowledge across Learning Processes"
_ICLR.cc/2019/Conference_

### Official Review · AnonReviewer2 · 2018-11-01
**Interesting idea, sufficient empirical evidence, with certain questionable tricks**

**Rating:** 8
**Confidence:** 3

**Review:**

This paper proposes Leap, a meta-learning procedure that finds better initialization for new tasks. Leap is based on past training/optimization trajectories and updates the initialization to minimize the total trajectory lengths. Experiments show that Leap outperforms popular alternatives like MAML and Reptile.

Pros
- Novel idea
- Relatively well-written
- Sufficient experiment evidence

Cons
- There exist several gaps between the theory and the algorithm

I have several concerns.
1. The idea is clearly delivered, but there are several practical treatments that are questionable. The first special treatment is that on page 5, when the objective is increased instead of decreased, the sign of the f part is flipped, which is not theoretically sound. It is basically saying that when we move from psi^i to psi^{i+1} with increased objective, we lie to the meta-learner that it is decreasing. The optimization trajectory is what it is. It would be beneficial to see the effect of removing this trick, at least in the experiments. Second, replacing the Jacobian with the identity matrix is also questionable. Suppose we use a very small but constant learning rate alpha for a convex problem. Then J^i=(I-G)^i goes to the zero matrix as i increases (G is small positive). However, instead, the paper uses J^i=I for all i. This means that the contributions for all i are the same, which is unsubstantiated.

2. The proof of Thm1 in Appendix A is not complete. For example, "By assumption, beta is sufficiently small to satisfy F", which I do not understand the inequality. Is there a missing i superscript? Isn't this the exact inequality we are trying to prove for i=0? As another example, "if the right-most term is positive in expectation, we are done", how so? BTW, the right-most term is a vector so there must be something missing. It would be more understandable if the proof includes a high-level proof roadmap, and frequently reminds the reader where we are in the overall proof now.

3. The set \Theta is not very well-defined, and sometimes misleading. Above Eq.(6), \Theta is mathematically defined as the intersection of points whose final solutions are within a tolerance of the *global* optimum, which is in fact unknown. As a result, finding a good initialization in \Theta for all the tasks as in Eq.(5) is not well-defined.

4. About the experiments. What is the "Finetuning" in Table 1? Presumably it is multi-headed but it should be made explicit. What is the standard deviation for Fig.4? The claim that "Leap learns faster than a random initialization" for Breakout is not convincing at all.

Minors
- In Eq.(4), f is a scalar so abs should suffice. This also applies to subsequent formulations.
- \mu is introduced above Eq.(8) but never used in the gradient formula.
- On p6, there is a missing norm notation when introducing the Reptile algorithm.

---

> ### Author Response · Authors · 2018-11-19
> **Thank you for insightful comments (I/II)**
>
> We are grateful for your insightful comments and glad that you like many aspects of the paper. We understand that your concerns are related to some theoretical parts, so we hope that our clarifications below, extra experiments and appropriate amendments to the paper will resolve your concerns fully.
>
> > 1a) the sign of the f part is flipped which is not sound
>
> We believe that this concern comes from the fact that we omitted clarifying the role of this term which is only a practical regularizer. Hence we apologize and sympathize with your comment. To resolve your concern, let us first say that the regularizer is not essential but, rather, only an optional stabilizer that practically allows for the use of larger step sizes. We have added an ablation study (appendix C) where we show that the regularizer yields faster convergence in terms of meta gradient steps, however the final performance is largely equivalent.
>
> Our revised manuscript provides a more through motivation in (section 2.3) that we hope you agree with: in short, the motivation for the stabilizer is that in stochastic gradient descent, the gradient path can be rather volatile. As you say, the gradient path is what it is. As long as it converges, so will Leap (with or without the regularizer). But if we could reduce the noise inherent in SGD, Leap could converge faster, and the stabilizer is a heuristic to do that. Other heuristics can certainly be used, or none at all.
>
>
> > 1b) Second, replacing the Jacobian with the identity matrix is also questionable
>
> We have added a new ablation study (appendix B) which shows that the approximation is quite tight, even for relatively large learning rates. With our best performing (inner loop) learning rate, we find the approximation to be accurate to the fourth decimal. We hope that you will find this study satisfying, although we also hope that you appreciate that the question about what Jacobian approximation is better to use is out of scope of our paper and does not affect the main point of our work.
>
> More generally, any meta-learner that optimizes over the inner training process must approximate the Jacobian in order to scale, and the identity assumption is a frequently used approach that works well in practice. The purpose of this paper is to present a new way of framing meta-learning such that it can scale, leveraging existing approach to the approximations we must make. Our approach relies on prior work by Finn et. al. (2017), who found that the assumptions works well, and Nichol et. al. (2017) who found a similar empirical result, and further showed formally that detaching the Jacobians still optimizes the original objective (approximately).
>
> Importantly, we can control the precision of this approximation through the learning rate and the number of gradient steps: for any given number of gradient steps (yielding an upper bound on i), we can choose \alpha such to ensure the approximation is sufficiently accurate to allow meta learning. Our ablation study (appendix B) shows that the resulting restriction on \alpha is not severe.
>
> In summary, we believe that our (admittedly sub-optimal) treatment of the Jacobian is well motivated and in-line with existing methods; we do agree that it can be improved, but this is out of our paper's scope.
>
>
> > 2) The proof of Theorem 1 is not complete
>
> We sympathize with your concern and agree that the presentation of the proof can be made clearer. We have taken your suggestions into account and re-organized the proof to directly establish the desired result, d(\psi^0_{s+1}) < d(\psi^0_s), and have included several commentaries to ensure each step of the proof is clearly linked to the overall objective.
> As for your specific questions:
> \beta_s is assumed to be sufficiently small to allow for gradient descent on the current baseline. The proof needs to establish that a new baseline generated from the updated initialization has a shorted gradient path length.
> There was indeed a typo in the theorem, leading to a missed term (hence the vector).

---

> > ### Author Response · Authors · 2018-11-19
> > **Thank you for insightful comments (II/II)**
> >
> > > 3)  \Theta is not very well-defined
> >
> > We fully understand your sentiment, we think it’s caused by a misunderstanding stemming from the way we describe this constraint. We have updated the paper (section 2.2) to make the following explanations clearer. Intuitively, the purpose of \Theta is to provide an upper bound on what we, as modellers, consider as good performance. Mathematically, we characterize this as some \epsilon bound on the global optimum. However, the only relevant bound is the level of performance we could achieve through our second-best option, i.e. starting from a random initialization or from fine-tuning. This level of performance is what \Theta is about.  As such, the global minimum is redundant in the definition, and we have revised the definition of \Theta to avoid it, instead emphasizing that \Theta is defined by the performance we could otherwise achieve.
> >
> > > 4) Experiments
> >
> > Thank you for these comments. We were aware of the need for multiple seeds for the RL experiments have updated our results with averages over 10 seeds. Notably, we find that Leap outperforms a random initialization because it more consistently finds good exploration spaces.
> >
> > Please also note that we have made further additions to our experimental section (as per our top-level reply) as requested by other reviewers.

---

> > > ### Comment · AnonReviewer2 · 2018-11-30
> > > **Thank you for the updates and clarifications**
> > >
> > > The manuscript has been improved substantially thus I updated my score.
> > >
> > > 1. On page 6, there is still no explicit formula to show how \mu (the stabilizer) is applied to the meta-gradient.
> > >
> > > 2. In Appendix B, what is the ||_1 norm on the Jacobian? We need to be clear about matrix norms, because _1 can mean Schatten 1-norm, vector-induced 1-norm etc.
> > >
> > > Minor
> > > - As in Algo.1, the meta-gradient is applied to theta not psi, so it would make more sense for Thm.1 (and the proof in Appendix A) to use theta instead of psi (also to avoid potential confusion).
> > > - Correct me if I am wrong, for general p, in the the meta-gradient (Eq.8), the last term should have a single exponent (p-2) on the L_2 norm instead of p(p-2). Moreover, the coefficient before the expectation should be p instead of 2 (this does not affect the algorithm though since we have \beta to control step size).
> > > - In Appendix B, the first equation, \alpha^{i^2} is misleading, maybe use (\alpha^i)^2
> > > - In Appendix A, right before "with p = 2 defining...", there is a \psi^0_{s,+1} that should be \psi^0_{s+1},

---

> > > > ### Author Response · Authors · 2018-12-06
> > > > **Thank you for your update**
> > > >
> > > > Dear reviewer,
> > > >
> > > > Thank you for taking the time to consider our rebuttal and revised manuscript.
> > > >
> > > > You raise good points and we will address these in a final version of the paper; we have added a sentence following the stabilizer describing how it affects the meta gradient, and to answer your question about the norm in the Jacobian approximation, it is indeed the Schatten 1-norm.

---

### Official Review · AnonReviewer1 · 2018-11-03
**Review of Transferring Knowledge across Learning Processes**

**Rating:** 8
**Confidence:** 4

**Review:**

\documentclass[10pt]{article}
\usepackage{geometry}[1in]
\usepackage{amsfonts}
\usepackage{amssymb}
\usepackage{amsmath}
\usepackage{enumerate}
\usepackage{indentfirst}

\begin{document}

	\section*{SUMMARY}

	The article proposes Leap, a novel meta-learning objective aimed at outperforming state-of-the-art approaches when dealing with collections of tasks that exhibit substantial between-task diversity.

	Similarly to prior work such as MAML [1] or Reptile [2], the goal of Leap is to learn an initialization $\theta_{0}$ for the model parameters, shared across tasks, which leads to good and data-efficient generalization performance when fine-tuning the model on a set of held-out tasks. In a nutshell, what sets Leap apart from MAML or Reptile is its cost function, which explicitly accounts for the entire path traversed by the model parameters during task-specific fine-tuning -- i.e., ``inner loop'' optimization --, rather than mainly focusing on the final value attained by the model parameters after fine-tuning. More precisely, Leap looks for an initialization $\theta_{0}$ of the model parameters such that the energy of the path traversed by $\gamma_{\tau}(\theta) = (\theta, f_{\tau}(\theta))$ while fine-tuning $\theta$ to optimize the loss $f_{\tau}(\theta)$ of a task $\tau$ is minimized, on average, across $\tau \sim p(\tau)$. Thus, it could be argued that Leap extends Reptile, which can be informally understood as seeking an initialization $\theta_{0}$ that minimizes the average squared Euclidean distance between $\theta_{0}$ and the model parameters after fine-tuning on each task $\tau \sim p(\tau)$ [2, Section 5.2], by using a distance function between initial and final model parameters that accounts for the geometry of the loss surface of each task during optimization.

	The final algorithm introduced in the paper considers however a variant of the aforementioned cost function, motivated by its authors on the basis of stabilising learning and eliminating the need for Hessian-vector products. The resulting approach is then evaluated on image recognition tasks (Omniglot plus a set of six additional computer vision datasets) as well as reinforcement learning tasks (Atari games).

	\section*{HIGH-LEVEL ASSESSMENT}

	The article proposes an interesting extension of existing work in meta-learning. In a slightly different context (meta-optimization), recent work [3] pointed out the existence of a ``short-horizon bias'' which could arise when using meta-learning objectives that apply only a small number of updates during ``inner-loop'' optimization. This observation is well-aligned with the motivation of this article, in which the authors attempt to complement successful methods like MAML or Reptile to perform well also in situations where a large number of gradient descent-based updates are applied during task-specific fine-tuning. Consequently, I believe the article is timely and relevant.

	Unfortunately, I have some concerns with the current version of the manuscript regarding (i) the proposed approach and the way it is motivated, (ii) the underlying theoretical results and, perhaps most importantly, (iii) the experimental evaluation. In my opinion, these should ideally be tackled prior to publication. Nonetheless, I believe that the proposed approach is promising and that these concerns can be either addressed or clarified. Thus I look forward to the rebuttal.

	\section*{MAJOR POINTS}

	\subsection*{1. Issues regarding proposed approach and its motivation/derivation}

	\textbf{1.a} Section 2.1 argues in favour of studying the path traversed by $\gamma_{\tau}(\theta) = (\theta, f_{\tau}(\theta))$ rather than the path traversed by the model parameters $\theta$ alone. However, this could in turn exacerbate the difficulty in dealing with collections of tasks for which the loss functions have highly diverse scales. For instance, taking the situation to the extreme, one could define an equivalence class of tasks $[\tau] = \left\{\tau \mid f_{\tau}(\theta) = g(\theta) + \mathrm{constant} \right\}$ such that any two tasks $\tau_{1}, \tau_{2} \in [\tau]$ would essentially represent the same underlying task, but could lead to arbitrarily different values of the Leap cost function.

	Given that Leap is a model-agnostic approach, like MAML or Reptile, and thus could be potentially applied in many different settings and domains, I believe the authors should study and discuss (theoretically or experimentally) the robustness of Leap with respect to between-task variation in the scale of the loss functions and, in case the method is indeed sensitive to those, propose an effective scheme to normalize them.

	\textbf{1.b} The current version of the manuscript motivates defining the cost function in terms of $\gamma_{\tau}(\theta) = (\theta, f_{\tau}(\theta))$ rather than the model parameters $\theta$ alone in order to ``avoid information loss'', making it seem that this modification is ``optional'' or, at least, not critical. Nevertheless, taking a closer look at the Leap objective and the meta-updates it induces, I believe it might actually be essential for the correctness of the approach. I elaborate this view in what follows. Let us write the Leap objective for a task $\tau$ as
	\[
	F_{\tau}(\theta_{0},\widetilde{\theta}_{0}) = \underbrace{\sum_{i=0}^{K_{\tau} - 1}{\left\vert\left\vert u^{(i+1)}_{\tau}(\widetilde{\theta}_{0}) - u^{(i)}_{\tau}(\theta_{0}) \right\vert\right\vert^{2}}}_{C_{\tau, 1}(\theta_{0},\widetilde{\theta}_{0})} + \underbrace{\sum_{i=0}^{K_{\tau} - 1}{\left( f_{\tau}\left(u^{(i+1)}_{\tau}(\widetilde{\theta}_{0})\right) - f_{\tau}\left(u^{(i)}_{\tau}(\theta_{0})\right) \right)^{2}}}_{C_{\tau, 2}(\theta_{0},\widetilde{\theta}_{0})},
	\]
	where $\widetilde{\theta}_{0}$ denotes a ``frozen'' or ``detached'' copy of $\theta_{0}$ and $u^{(i)}_{\tau}$ maps $\theta_{0}$ to $\theta_{i}$, the model parameters after applying $i$ gradient descent updates to $f_{\tau}$ according to Equation (1) in the manuscript. Then, differentiating $C_{\tau, 1}(\theta_{0},\widetilde{\theta}_{0})$ and $C_{\tau, 2}(\theta_{0},\widetilde{\theta}_{0})$ with respect to $\theta_{0}$ separately yields:
	\begin{align*}
	\nabla_{\theta_{0}} C_{\tau, 1}(\theta_{0},\widetilde{\theta}_{0}) &= -2 \sum_{i=0}^{K_{\tau} - 1}{J_{i}^{T}\left(\theta_{i+1} - \theta_{i} \right)} = -2 \alpha \sum_{i=0}^{K_{\tau} - 1}{J_{i}^{T} g_{i}} \\
	\nabla_{\theta_{0}} C_{\tau, 2}(\theta_{0},\widetilde{\theta}_{0}) &= -2 \sum_{i=0}^{K_{\tau} - 1}{\left(f_{\tau}(\theta_{i+1}) -  f_{\tau}(\theta_{i})\right) J_{i}^{T}g_{i}} = -2 \sum_{i=0}^{K_{\tau} - 1}{\Delta f^{i}_{\tau} J_{i}^{T}g_{i}}
	\end{align*}
	where $J_{i} = J_{\theta_{0}}u^{(i)}_{\tau}(\theta_{0})$ denotes the Jacobian of $u^{(i)}_{\tau}$ with respect to $\theta_{0}$, $g_{i} = \left. \nabla_{\theta} f_{\tau}(\theta)\right\rvert_{\theta=\theta_{i}}$ denotes the gradient of the loss function $f_{\tau}$ evaluated at $\theta_{i}$ and $\Delta f^{i}_{\tau} = f_{\tau}(\theta_{i+1}) -  f_{\tau}(\theta_{i})$ stands for the change in the loss function after the $i$-th update. To simplify the exposition, a constant ``inner-loop'' learning rate and no preconditioning were assumed, i.e., $\alpha_{i} = \alpha$ and $S_{i} = I$.

	Furthermore, the article claims that all Jacobian terms are approximated by identity matrices (i.e., $J_{i} = I$) as suggested in Section 5.2 of [1], leading to the following approximations:
	\begin{align*}
		\nabla_{\theta_{0}} C_{\tau, 1}(\theta_{0},\widetilde{\theta}_{0}) \approx -2 \alpha \sum_{i=0}^{K_{\tau} - 1}{ g_{i}} \\
		\nabla_{\theta_{0}} C_{\tau, 2}(\theta_{0},\widetilde{\theta}_{0}) \approx -2 \sum_{i=0}^{K_{\tau} - 1}{\Delta f^{i}_{\tau} g_{i}}
	\end{align*}

	Interestingly, it can be seen that the contribution to the meta-update of the energy of the path traversed by the model parameters $\theta$, $g_{\mathrm{Leap},1} =\nabla_{\theta_{0}} C_{\tau, 1}(\theta_{0},\widetilde{\theta}_{0})$, actually points in exactly the opposite direction than the meta-update of Reptile, given by $g_{\mathrm{Reptile}} = \sum_{i=0}^{K_{\tau} - 1}{g_{i}}$ (e.g. Equation (27) in [2]). In summary, if the Leap objective was defined in terms of $\theta$ rather than $(\theta, f_{\tau}(\theta))$, minimising the Leap cost function should maximise Reptile's cost function and viceversa. It is only the term $g_{\mathrm{Leap},2} =\nabla_{\theta_{0}} C_{\tau, 2}(\theta_{0},\widetilde{\theta}_{0})$ that presumably ``re-aligns'' $g_{\mathrm{Reptile}}$ and $g_{\mathrm{Leap}} = g_{\mathrm{Leap},1} + g_{\mathrm{Leap},2}$. Indeed,
	\[
	g_{\mathrm{Leap}} = 2 \sum_{i=0}^{K_{\tau} - 1}{\left(-\Delta f^{i}_{\tau} - \alpha \right) g_{i}}
	\]
	will have positive inner product with $g_{\mathrm{Reptile}}$ if each gradient update yields a sufficient decrease in the loss $f_{\tau}$, that is, $\Delta f^{i}_{\tau} < -\alpha$.

	Moreover, I also wonder if this is the reason why the authors introduce the ``regularization'' term $\mu_{\tau}^{i}$, which as it currently stands in the manuscript, does not seem to relate in a particularly intuitive manner to the original objective of minimising the energy of $\gamma(t)$. By introducing $\mu_{\tau}^{i}$, the term $C_{\tau, 2}(\theta_{0},\widetilde{\theta}_{0})$ becomes
	\[
		C^{\prime}_{\tau, 2}(\theta_{0},\widetilde{\theta}_{0}) = \sum_{i=0}^{K_{\tau} - 1}{-\mathrm{sign}  \left( f_{\tau}\left(u^{(i+1)}_{\tau}(\widetilde{\theta}_{0})\right) - f_{\tau}\left(u^{(i)}_{\tau}(\theta_{0})\right) \right) \left( f_{\tau}\left(u^{(i+1)}_{\tau}(\widetilde{\theta}_{0})\right) - f_{\tau}\left(u^{(i)}_{\tau}(\theta_{0})\right) \right)^{2}},
	\]
	leading to $g^{\prime}_{\mathrm{Leap},2} = 2 \sum_{i=0}^{K_{\tau} - 1}{\vert \Delta f^{i}_{\tau} \vert g_{i}}$ and
	\[
	g^{\prime}_{\mathrm{Leap}} = 2 \sum_{i=0}^{K_{\tau} - 1}{\left(\vert \Delta f^{i}_{\tau} \vert - \alpha \right) g_{i}}.
	\]
	In turn, this relaxes the sufficient condition under which Leap and Reptile lead to meta-updates with positive inner product, namely, it changes the condition $\Delta f^{i}_{\tau} < -\alpha$ by a less restrictive counterpart $\vert \Delta f^{i}_{\tau} \vert \ge \alpha$.

	If these derivations happen to be correct, then I believe the way Leap is currently motivated in the article could be argued to be slightly misleading. What seems to be its main inspiration, accounting for the path that the model parameters traverse during fine-tuning, does not seem to be what drives the meta-updates towards the ``correct'' direction. Instead, the component of the objective due to the path traversed by the loss function values appears to be more important or, at least, not optional. Furthermore, I believe the regularization term $\mu_{\tau}^{i}$ should be better motivated, as the current version of the manuscript does not seem to justify its need clearly enough.

	Finally, under the assumption that the above is not mistaken, I wonder whether further tweaks to the meta-update, such as $g^{\prime\prime}_{\mathrm{Leap}} = 2 \sum_{i=0}^{K_{\tau} - 1}{\mathrm{max}\left(\vert \Delta f^{i}_{\tau} \vert - \alpha, 0 \right) g_{i}}$, could perhaps turn out to be helpful as well.

	\subsection*{2. Theoretical results}

	\textbf{2.a} Theorem 1 currently claims that the Pull-Forward algorithm converges to a local minimum of Equation (5). However, due to the non-convexity of the objective function, only convergence to a stationary point is established.

	\textbf{2.b} Most importantly, I am not entirely certain that the proof of Theorem 1 is complete in its current form. As I understand it, using the notation introduced by the authors in Appendix A, the following identities hold:
	\begin{align*}
		F(\psi_{s};\Psi_{s}) &= \mathbb{E}_{\tau,i} \vert\vert h_{\tau}^{i} - z_{\tau}^{i} \vert\vert^{2} \\
		F(\psi_{s+1};\Psi_{s}) &= \mathbb{E}_{\tau,i} \vert\vert h_{\tau}^{i} - x_{\tau}^{i} \vert\vert^{2} \\
		F(\psi_{s};\Psi_{s+1}) &= \mathbb{E}_{\tau,i} \vert\vert y_{\tau}^{i} - z_{\tau}^{i} \vert\vert^{2} \\
		F(\psi_{s+1};\Psi_{s+1}) &= \mathbb{E}_{\tau,i} \vert\vert y_{\tau}^{i} - x_{\tau}^{i} \vert\vert^{2}.
	\end{align*}

	The bulk of the proof is then devoted to show that $\mathbb{E}_{\tau,i} \vert\vert y_{\tau}^{i} - z_{\tau}^{i} \vert\vert^{2} = F(\psi_{s};\Psi_{s+1}) \ge \mathbb{E}_{\tau,i} \vert\vert y_{\tau}^{i} - x_{\tau}^{i} \vert\vert^{2} = F(\psi_{s+1};\Psi_{s+1})$. However, I do not immediately see how to make the final ``leap'' from $F(\psi_{s+1};\Psi_{s+1}) \le F(\psi_{s};\Psi_{s+1})$ to the actual claim of the Theorem, $F(\psi_{s+1};\Psi_{s+1}) \le F(\psi_{s};\Psi_{s})$.

	\subsection*{3. Experimental evaluation}

	\textbf{3.a} The experimental setup of Section 4.1 closely resembles experiments described in articles that introduced continual learning approaches, such as [4]. However, rather than including [4] as a baseline, the current manuscript compares against meta-learning approaches typically used for few-shot learning, such as MAML and Reptile. Consequently, I would argue the combination of experimental setup and selection of baselines is not entirely fair or, at least, it is incomplete.

	To this end, I would suggest to (i) include [4] (or a related continual learning approach) as an additional baseline in the experiments currently described in Section 4.1 as well as (ii) perform a new experiment to compare the performance of Leap to that of MAML and Reptile in few-shot classification tasks using OmniGlot and/or Mini-ImageNet as datasets.

	\textbf{3.b} The Multi-CV experiment described in Section 4.2 currently does not have strong baselines other than Leap. If possible, I would suggest including [5] in the comparison, as it is the article which inspired this particular experiment.

	\textbf{3.b} Likewise, the same holds for the experiment described in Section 4.3. In this case, I would suggest comparing to [4] for the same reason described above.

	\section*{MINOR POINTS}

	\begin{enumerate}

	\item In Section 2.1, it is claimed that "gradients that largely point in the same direction indicate a convex loss surface, whereas gradients with frequently opposing directions indicate an ill-conditioned loss landscape". Nevertheless, convex loss surfaces can in principle be ill-conditioned as well.

	\item Introducing a mathematical definition for the metric "area under the training curve" could make the experiment in Section 4.1 more self-contained.

	\item Several references are outdated, as they cite preprints that have since been accepted at peer-reviewed venues.

	\item The reinforcement learning experiments in Section 4.3 would benefit from additional runs with multiple seeds, and the subsequent inclusion of confidence intervals.

	\item I believe certain additional experiments could be insightful. For example, (i) studying how sensitive the performance of Leap is to parameter of the ``inner-loop'' optimizer (e.g. choice of
	optimizer, learning rate, batch size) or (ii) describing how the introduction of $\mu_{\tau}^{i}$ affects the performance of Leap.

	\end{enumerate}

	\section*{TYPOS}

	\begin{enumerate}

	\item The first sentence entirely in page 6 appears to have a superfluous word.

	\item The Taylor series expansion in the proof of Theorem 1 is missing the $O(\bullet)$ terms (or a $\approx$ sign).

	\item Also in the proof of Theorem 1, if $c_{\tau}^{i} = (\delta_{\tau}^{i})^{2} - \alpha_{\tau}^{i}\xi_{\tau}^{i}\delta_{\tau}^{i}$, wouldn't $\omega = \underset{\tau, i}{\mathrm{sup}} \langle \hat{x}^{i}_{\tau} - \hat{z}^{i}_{\tau}, g(\hat{x}^{i}_{\tau}) - g(\hat{z}^{i}_{\tau})\rangle + \xi_{\tau}^{i}\delta_{\tau}^{i}$ instead?

	\end{enumerate}

        \section*{ANSWER TO REBUTTAL}
        Please see comments in the thread.


	\section*{REFERENCES}

	\begin{enumerate}[ {[}1{]} ]
		\item Finn et al. ``Model-Agnostic Meta-Learning for Fast Adaptation of Deep Networks.'' International Conference on Machine Learning. 2017.
		\item Nichol et al. ``On First-Order Meta-Learning Algorithms.'' arXiv preprint. 2018
		\item Wu et al. ``Understanding Short-Horizon Bias in Stochastic Meta-Optimization.'' International Conference on Learning Representations. 2018.
		\item Schwarz et al. ``Progress \& Compress: A scalable framework for continual learning.''  International Conference on Machine Learning. 2018.
		\item Serr{\`a} et al. ``Overcoming Catastrophic Forgetting with Hard Attention to the Task.''  International Conference on Machine Learning. 2018.
	\end{enumerate}
\end{document}

---

> ### Author Response · Authors · 2018-11-19
> **Thank you for a thorough review! (I/II)**
>
> Thank you for such a thorough review! We are very grateful for your feedback and are excited to use it to improve our paper. Together with our added clarifications and new experiments, we hope that we now address your concerns in full. If not, we are looking forward to discuss more. Please see details below.
>
> > 1a) including the loss in the task manifold can make learning unstable if tasks have losses on different magnitudes.
>
> Leap is indeed sensitive to differing scales across task objective functions. However, this sensitivity is not due to incorporating the loss in the task manifold, and would exist even if it were omitted. It arises from the fact that the meta gradient is an average over task gradients, which gives tasks with larger gradients (on average) greater influence. As such, this is a general problem applying equally to similar methods, like MAML and Reptile.
>
> We were aware of this and after submission we have experimented with formulations that alleviate this issue. In fact, using the approximate gradient path length (as opposed to the energy) yields a meta gradient that scales all task gradients by a task specific norm that avoids this issue. This is an important improvement, and we are grateful for your insight here. We have generalized Leap (section 2) to allow both for a meta learning objective under the energy metric as well as under the length metric. In appendix C, we have added a new thorough ablation study across design choices and find that while Leap converges faster under the length metric (in terms of meta training steps), final performance is equivalent.
>
> > 1b) Including the loss in the task manifold is not optional, as suggested by the paper, but essential, to produce loss-minimizing meta-gradients.
>
> We are very grateful for the time you have taken to investigate this issue! Unfortunately, your argument is based on an incorrect derivation, as there is a small mistake in the second inequality on C_1: you replace \theta_{i+1} - \theta_i with \alpha g_i, but that would imply gradient ascent. The right identity is \theta_{i+1} - \theta_i = - \alpha g_i (see eq. 1).
>
> Correcting for this, C_1 is not only aligned with the Reptile gradient, it *is* the reptile gradient (this exact equivalence breaks down when we use length metric as meta objective, or if we were to jointly learn other aspects of the gradient update, e.g. learning rate / preconditioning).
>
> Our newly added ablation study in appendix C shows that Leap can converge even if we remove the loss from the task manifold, but does so at a significantly slower rate and learns a less useful initialization. Including the loss is a key feature of our framework, because it tells Leap how impactful a gradient step is: a gradient step that has a large influence on the loss will be given greater attention, allowing Leap to “prioritize”. The importance of this information is clearly illustrated in the Omniglot experiment, were Leap does significantly better than Reptile.
>
> This ability to prioritize is also what motivated us in adding a regularizer, which perhaps is better called a stabilizer. Leap prioritizes large loss deltas, so if the learning rate is too large, or the gradient estimator very noisy, it could happen that we get a large increase in the loss, which would then be prioritized by Leap. Being an anomaly, this doesn’t derail Leap–in the end, Leap follows the entire gradient path (see appendix C). As such, the stabilizer is not critical, but it does speed up training and allows the use more aggressive learning rates. Finally, as you point out, our formulation is just one heuristic, others may be better.

---

> > ### Author Response · Authors · 2018-11-19
> > **Thank you for a thorough review! (II/II)**
> >
> > > 2a) Theorem 1 only assert convergence to a stationary point
> >
> > Correct, apologies for our imprecision: we have updated the paper to reflect that Leap converges to a limit point in \Theta. Our point was that gradient descent on the pull-forward objective is equivalent to gradient descent on the original objective, which we now state explicitly in section 2.3.
> >
> > > 2b) The proof of Theorem 1 may be incomplete
> >
> > The final “leap” is implicit and unfortunately not clearly explained. We have substantially re-organized the proof to prove the desired inequality, d(\psi^0_{s+1}) < d(\psi^0_s), directly. We have also added a commentaries to more clearly explain each step of the proof, to avoid any confusion as to what is being established.
> >
> > > 3) Unfair comparisons / lack of baselines
> >
> > To address concerns about lacking strong baselines, we have added two baseline related methods that do not regularize with respect to previous tasks, HAT (Serra et al., 2019)  and Progressive Nets (Rusu et al., 2017), to the Multi-CV experiment. We found neither HAT nor Progressive Nets match Leap’s performance.
> >
> > We hope that you appreciate that the continual learning problem is very different from the type of multitask learning we are considering here. The point we are trying to make in our paper is that MAML, Reptile, and similar methods cannot scale to problems that require more than a handful of gradient steps, while Leap can. As such, we believe that treating Omniglot–a standard few-shot learning problem where meta learning does well– as a multi-shot learning problem is highly relevant. We are not arguing that Leap is superior at few-shot learning, though it could be.
> >
> > Please also note that we have made further additions to our experimental section (as per our top-level reply) as requested by other reviewers.

---

> > > ### Comment · AnonReviewer1 · 2018-11-23
> > > **Thank you for taking the time to address the review in great detail (I/II)**
> > >
> > > # HIGH-LEVEL ASSESSMENT (UPDATED)
> > >
> > > After reading the author rebuttal and going through the revised manuscript, I believe the authors have successfully addressed the vast majority of concerns I had about the original version of the paper.
> > >
> > > Based on the current version of the article, I lean strongly towards acceptance and have modified my score accordingly.
> > >
> > > # STATE OF PREVIOUSLY RAISED MAJOR POINTS
> > >
> > > 1. In my original review, I raised issues regarding the way LEAP was motivated and derived; an opinion also voiced by Reviewer 2.
> > >
> > > I believe Section 2 of the revised manuscript has greatly improved in terms of clarity while simultaneously being more general.
> > >
> > > I apologise for the mistaken sign in $\Delta \theta_{\tau}^{i}$ in the subsequent analysis. In hindsight, I should have definitely caught that error based on the very unintuitive conclusions that ensue!. The fact that LEAP reduces to Reptile when minimising the expected energy of the "non-augmented" gradient flow makes perfect sense and helps understand what LEAP's "place" is alongside MAML and Reptile.
> > >
> > > The authors have also extended LEAP to minimise either the length or the energy of the gradient path, rather than minimising only the energy. This possibility was loosely mentioned in the original manuscript, but not implemented. As pointed out in their rebuttal, minimising the length of the gradient path instead of the energy implicitly "normalises" the magnitude of the gradient w.r.t. the initialisation $\theta_{0}$ across tasks (Eq. 8), which might make LEAP most robust against heterogeneity in the scale of task losses.
> > >
> > > The new ablation studies included in Sections B and C of the Appendix are also a great addition to study/justify empirically some of the more heuristic aspects of the paper.
> > >
> > > 2. The original review also raised some concerns regarding Theorem 1 and its proof; a point also raised by Reviewer 2.
> > >
> > > The statement of Theorem 1 and, most importantly, its proof, have been almost entirely rewritten. To the best of my knowledge, I believe the revised version is correct (potential minor inconsequential caveats described below), and is now much clearer and easy to follow.
> > >
> > > 3. Besides carrying out the new ablation studies, the authors have introduced two additional baselines in Section 4.2 and now report aggregated results for 10 different seeds in Section 4.3.
> > >
> > > I still believe that having included additional baselines also in Sections 4.1 and 4.3, as well as evaluating LEAP in a "less favourable" few-shot learning scenario, could have further strengthened the paper. Nevertheless, given the time (and possibly compute) constraints, the revised manuscript also improved considerably in terms of experimental results and, most importantly, already provides sufficient evidence that LEAP can outperform existing approaches when tasks are sufficiently diverse.

---

> > > > ### Comment · AnonReviewer1 · 2018-11-23
> > > > **Thank you for taking the time to address the review in great detail (II/II)**
> > > >
> > > > # MINOR POINTS ABOUT THE REVISED MANUSCRIPT
> > > >
> > > > 1. The claim that "if both tasks have convex loss surfaces there is a unique optimal initialization that achieves Pareto optimality in terms o total path distance", while true, might not be so helpful, since initialization should in theory be irrelevant for convex losses.
> > > >
> > > > 2. Currently, Eq. 8 is derived assuming the stabilizer $\mu$ is included in the loss. However, the stabilizer is only introduced afterwards. This might be confusing for some readers if they attempt to derive Eq. 8 themselves when they first encounter it, prior to finishing reading page 6 entirely.
> > > >
> > > > 3. While I think the role of the stabilizer heuristic is much more clearly explained now, there is a claim that still confuses me slightly. In the last paragraph of page 6, it is said that "The stabilizer ... reduces the weight placed on the gradient of $f_{\tau}(\theta_{\tau}^{i})$". However, under the simplifying assumption $S_{\tau}^{i} = I$, one would have $g_{i} := \nabla f_{\tau}(\theta_{\tau}^{i})$ and $\Delta \theta_{\tau}^{i} = -\alpha_{\tau}^{i} g_{i}$. Then, without stabilizer, the "weight" of $g_{i}$ would be $\alpha_{\tau}^{i} - \Delta f_{\tau}^{i}$ while with stabilizer, the "weight" of $g_{i}$ would then be $\alpha_{\tau}^{i} + \vert \Delta f_{\tau}^{i} \vert$, which in principle could be larger  (in magnitude) than the weight without stabilizer. Nonetheless, if this is not mistaken, it would be clear that the stabilizer ensures $g_{i}$ is never effectively followed in the ascent direction in rare cases when $\Delta f_{\tau}^{i}$ is large and positive.
> > > >
> > > > 4. I believe there might be a small mistake in the proof of Theorem 1. Nevertheless, even if this were the case, I think it would not affect the conclusion.
> > > >
> > > > In the middle of page 15, a derivation implies that $\langle h_{\tau}^{i} -  z_{\tau}^{i}, z_{\tau}^{i} -  x_{\tau}^{i} \rangle = -\alpha_{\tau}^{i} \langle g(z_{\tau}^{i}), z_{\tau}^{i} -  x_{\tau}^{i}\rangle$. However, I believe this ignores the contribution of the extra dimension corresponding to the loss function values. That is, $\langle h_{\tau}^{i} -  z_{\tau}^{i}, z_{\tau}^{i} -  x_{\tau}^{i} \rangle = \langle \hat{h}_{\tau}^{i} -  \hat{z}_{\tau}^{i}, \hat{z}_{\tau}^{i} -  \hat{x}_{\tau}^{i} \rangle + \left(f_{\tau}(\hat{h}_{\tau}^{i}) - f_{\tau}(\hat{z}_{\tau}^{i})\right)\left(f_{\tau}(\hat{z}_{\tau}^{i}) - f_{\tau}(\hat{x}_{\tau}^{i})\right)$. Nevertheless, I think that using $\langle \hat{h}_{\tau}^{i} -  \hat{z}_{\tau}^{i}, \hat{z}_{\tau}^{i} -  \hat{x}_{\tau}^{i} \rangle = -\alpha_{\tau}^{i} \langle g(\hat{z}_{\tau}^{i}), \hat{z}_{\tau}^{i} -  \hat{x}_{\tau}^{i}\rangle$ and $\left(f_{\tau}(\hat{h}_{\tau}^{i}) - f_{\tau}(\hat{z}_{\tau}^{i})\right)\left(f_{\tau}(\hat{z}_{\tau}^{i}) - f_{\tau}(\hat{x}_{\tau}^{i})\right) = \left(-\alpha_{\tau}^{i} {\nabla f_{\tau}^{i}(\hat{z}_{\tau}^{i})}^{T} g(\hat{z}_{\tau}^{i}) + O(\alpha_{\tau}^{i})\right) \left(f_{\tau}(\hat{z}_{\tau}^{i}) - f_{\tau}(\hat{x}_{\tau}^{i})\right)$ should still allow bounding $\alpha_{\tau}^{i}$ from above to ensure the objective function decreases.
> > > >
> > > > I believe a similar issue (the contribution of the extra dimension not being explicitly shown) might also have occurred in the last step of the proof, when bounding $\vert\vert h_{\tau}^{i} - z_{\tau}^{i} \vert\vert^{p}$ from above by ${\alpha_{\tau}^{i}}^{p} \vert\vert g(\hat{z}_{\tau}^{i}) \vert\vert^{p}$. But as with the previous case, I don't think this would affect the actual argument being made.
> > > >
> > > >
> > > > # TYPOS
> > > >
> > > > Page 2:
> > > >
> > > > "... our framework can be _extend_ to learn ..."
> > > > "... initialization. _Differences_ schemes represent ..."
> > > >
> > > > Page 4:
> > > >
> > > > "... Leap converges on _an_ locally Pareto optimal ..."
> > > > "... and _progress_ via ..."
> > > >
> > > > Page 5:
> > > >
> > > > "... and _construct_ baseline gradient ... "
> > > >
> > > > Pages 8 and 9:
> > > >
> > > > length metric ($d_{2}$) and energy metric ($d_{1}$) -> length metric ($d_{1}$) and energy metric ($d_{2}$)
> > > >
> > > > Page 9:
> > > >
> > > > "27 games that _has_ an action space ..."
> > > >
> > > > Pages 9 and 19 (Tables 1 and 3):
> > > >
> > > > No pre-training AUC for the Facescrub task in bold, but the value for PNs is smaller.
> > > >
> > > > Page 18:
> > > >
> > > > "... (until _convergenve_) ..."

---

> > > > > ### Author Response · Authors · 2018-11-23
> > > > > **Thank you for your diligence**
> > > > >
> > > > > We are very impressed with your diligence and grateful for your input – your comments are very helpful in improving our manuscript! We are further grateful for your willingness to engage in the rebuttal and revise your review. Please see below for responses to your comments and questions.
> > > > >
> > > > > >  Further baselines on Omniglot, miniImagenet would strengthen the paper
> > > > >
> > > > > We respect your position, and given time and resources we would have been happy to oblige. We respectfully disagree with regards to miniImagenet being “less favourable” to Leap. Since Reptile outperforms MAML on miniImagenet, and Leap can be reduced to Reptile, Leap’s performance is “lower bounded” by Reptile. While we take your point that it would be interesting to see how much of a boost other configurations could provide in a few-shot setting, given the feedback we received, we chose to prioritize other parts of the paper as we felt that would add more value.
> > > > >
> > > > > >  Pareto optimality on convex loss surfaces
> > > > >
> > > > > We have found that giving people a visual crutch helps them to understand how Leap behaves. More generally, Leap converges to a locally Pareto optimal point. As a property though, we agree that it’s not particularly interesting, which is why we don’t emphasize it in the manuscript.
> > > > >
> > > > > > the stabilizer is still in the meta-gradient
> > > > >
> > > > > We noticed this as well and have fixed it: we agree that it should not be part of equation 8.
> > > > >
> > > > > > confusing claim: “the stabilizer reduces emphasis on the gradient of f(\theta)”
> > > > >
> > > > > Apologies for the confusion, our choice of words was somewhat unfortunate. We have revised the manuscript to clarify this point. The weight placed on the task gradient is indeed larger, but as you note, \mu guarantees we follow the descent direction. What we meant here is that \mu reduces the weight placed on following that anomalous line segment, instead attempting to avoiding that neighborhood it in the updated gradient path.
> > > > >
> > > > > > I believe there might be a small mistake in the proof of Theorem 1. Nevertheless, even if this were the case, I think it would not affect the conclusion.
> > > > >
> > > > > Thank you for pointing this out, we overloaded the definition of g (note the use of g(z), as opposed to g(^z) in the inner product). We have removed this overloading and explicitly define g(z), largely as you propose.
> > > > >
> > > > > Thank you for a careful read, all typos fixed.

---

### Official Review · AnonReviewer3 · 2018-11-03
**A new transfer learning method for knowledge transfer between distinct tasks**

**Rating:** 6
**Confidence:** 4

**Review:**

In this paper, the authors study an important transfer learning problem, i.e., knowledge transfer between distinct tasks, which is usually called 'far transfer' (instead of 'near transfer'). Specifically, the authors propose a lightweight framework called Leap, which aims to achieve knowledge transfer 'across learning processes'. In particular, a method for meta-learning (see Algorithm 1) is developed, which focuses on minimizing 'the expected length of the path' (see the corresponding term in Eqs.(4-6)). Empirical studies on three public datasets show the effectiveness of the proposed method. Overall, the paper is well presented.

Some comments/suggestions:
(i) The details of the experiments such as parameter configurations are missing, which makes the results not easy to be reproduced.

(ii) For the baseline methods used in the experiments, the authors are suggested to include more state-of-the-art transfer learning methods in order to make the results more convincing.

(iii) Finally, if the authors can use some commonly used datasets in existing transfer learning works, the comparative results will be more interesting.

---

> ### Author Response · Authors · 2018-11-19
> **Thank you for your review**
>
> We thank you for your review. We understand your sentiment and hope that our revised paper will alleviate any concern you may have. More specifically,
>
> > 1) The details of the experiments such as parameter configurations are missing
>
> Thank you for pointing out that further experimental details are needed. We will add further details to ensure our results are fully replicable during this week.
>
> > 2) Include more state-of-the-art transfer learning methods
>
> We have added results for Progressive Nets (Rusu et et., 2017), which is a rather demanding baseline as it has more than 8 times as many parameters as Leap, and HAT (Serra et al., 2018) whose paper inspired our setup. We find that they do not change any of our conclusions.
>
> > 3) use some commonly used datasets
>
> We would like to point out that all datasets used in our paper are frequent in transfer learning work of various kind; the point we are making here is that Leap is a general purpose framework that can tackle any of them. In particular, Omniglot is frequently used in few-shot learning (Vinyals et al., 2016, Snell et al., 2017, Finn et al., 2017, Nichol et al., 2017), while all datasets in the Multi-CV experiment are common in various forms of transfer learning (Serra et al., 2018, Zenke et al., 2018, Zagoruyko et al., 2017 (https://arxiv.org/abs/1612.03928)). Similarly, Atari is a notoriously difficult transfer learning problem (Schwarz et al., 2018, Rusu et. al., 2017).
>
> We appreciate the sentiment, and in an ideal world we would be happy to add further datasets and baselines to our experiments. However, given time and resource constraint, running multiple large-scale experiments is not feasible. In this paper, we chose Atari as our large-scale experiment. Please also note that we have made further additions to our experimental section (as per our top-level reply) as requested by other reviewers.

---

> ### Author Response · Authors · 2018-12-03
> **Following up on rebuttal**
>
> Dear reviewer,
>
> Following our rebuttal and discussion with R1 and R2, we hope that you find your main concerns addressed. Please let us know if there are any other questions we can answer.

---

### Author Response · Authors · 2018-11-19
**Summary of revisions in light of reviews**

[This is a top-level reply with only a summary of our changes, please see our answer to each individual reviewer thread for details]

Dear Reviewers, thank you for throughout and thoughtful feedback and for being overall positive about our work. We have worked through our manuscript and have made several additions (including new experiments) that clarifies the link between the theory and the algorithm, provides further insight into both, and significantly strengthens our experimental results. We hope these additions address any questions raised and address any concerns you may have. In particular, we have

Expanded section 2 to provide further insights into the framework and our proposed solution algorithm.

- Generalized Leap to allow for the use of either the energy or length metric as measure of gradient path distance.

- Re-organized the proof of theorem 1 to address concerns about completeness and clarity.

- Added ablation study with respect to (a) the inclusion of the loss in the task manifold, (b) the use of the energy or length metric, and (c) the use of a regularizer/stabilizer. In short, the more sophisticated the meta objective, the better Leap performs. The length metric converges faster, but final performance is largely equivalent. Adding the loss to the task manifold improves performance, while the stabilizer speeds up convergence.

- Added ablation study with respect to the Jacobian approximation, as a function of the learning rate. We find that we can use relatively large learning rates without significant deterioration of the approximation.

- Added HAT and Progressive Nets as baselines on Multi-CV. Neither of them outperforms Leap.

- Report confidence intervals on Atari games. We find that Leap does better than a random initialization by more consistently exploring useful parts of parameter space.

Please see answers to individual reviewer below, for particular comments.

---

### Author Response · Authors · 2018-11-23
**Summary of revisions 2 and 3**

Dear reviewers,

Please note that we have made minor revisions since our initial rebuttal, summarized below

- Revision 2 added details to experiments, as requested by R3
- Revision 3 fixed some typos, improved the explanation of the stabilizer, and addressed R1's comment wrt the proof

---

### Meta-Review · Area_Chair1 · 2018-12-13

**Confidence:** 4
**Recommendation:** Accept (Oral)

**Metareview:**

This paper proposes an approach for learning to transfer knowledge across multiple tasks. It develops a principled approach for an important problem in meta-learning (short horizon bias). Nearly all of the reviewer's concerns were addressed throughout the discussion phase. The main weakness is that the experimental settings are somewhat non-standard (i.e. the Omniglot protocol in the paper is not at all standard). I would encourage the authors to mention the discrepancies from more standard protocols in the paper, to inform the reader. The results are strong nonetheless, evaluating in settings where typical meta-learning algorithms would struggle. The reviewers and I all agree that the paper should be accepted, and I think it should be considered for an oral presentation.